

# Exact out-of-equilibrium steady states in the semiclassical limit of the interacting Bose gas

**Giuseppe Del Vecchio Del Vecchio[1*] Alvise Bastianello[2],**
**Andrea De Luca[3] and Giuseppe Mussardo[4]**

**1** Department of Mathematics, King's College London, Strand WC2R 2LS
**2** Institute for Theoretical Physics, University of Amsterdam,
Science Park 904, 1098 XH Amsterdam, The Netherlands
**3** Laboratoire de Physique Théorique et Modélisation (CNRS UMR 8089),
Université de Cergy-Pontoise, F-95302 Cergy-Pontoise, France
**4** SISSA and INFN, Sezione di Trieste, Via Bonomea 265, I-34136 Trieste, Italy

⋆ giuseppe.del_vecchio_del_vecchio@kcl.ac.uk

## Abstract

We study the out-of-equilibrium properties of a classical integrable non-relativistic theory, with a time evolution initially prepared with a finite energy density in the thermodynamic limit. The theory considered here is the Non-Linear Schrödinger equation which describes the dynamics of the one-dimensional interacting Bose gas in the regime of high occupation numbers. The main emphasis is on the determination of the late-time Generalised Gibbs Ensemble (GGE), which can be efficiently semi-numerically computed on arbitrary initial states, completely solving the famous quench problem in the classical regime. We take advantage of known results in the quantum model and the semiclassical limit to achieve new exact results for the momenta of the density operator on arbitrary GGEs, which we successfully compare with ab-initio numerical simulations. Furthermore, we determine the whole probability distribution of the density operator (full counting statistics), whose exact expression is still out of reach in the quantum model.



# 1 Introduction

The out-of-equilibrium physics is a fascinating subject which is currently at the center of an intense research activity: this field has witnessed a striking progress due to both new experimental techniques, mostly available in the world of cold atoms, and an impressive development of analytic and numerical methods (see, for instance, [1]). These advances have opened the door to explore novel phenomena and to answer fundamental inquiries in quantum mechanics and statistical physics, in particular about relaxation and equilibration in many-body systems. One of the most crucial achievements of the last years has been the possibility of experimentally realising almost-perfectly isolated quantum systems and, at the same time, tuning their interactions and the effective dimension so that one can investigate the *quantum quench* [2] in the lab, as done indeed in several experimental setups [3–18]. Arguably, the quantum quench is among the simplest out-of-equilibrium protocols: after the system is prepared in a given initial state, which could be pure or mixed, it is let evolve with a non-trivial time-independent Hamiltonian $\hat{H}$. As discussed below, one of the key topics of the out-of-equilibrium physics concerns with the nature of the late-time steady state which follows a quench, in particular the possibility to predict the asymptotic stationary values of the various observables of the system. In this paper we are going to address the following question: can we characterise the steady state which emerges in the infinite time limit of a *classical* integrable field theory? As we will show, the answer to this question proves to be remarkably rich and instructive, with several far-reaching consequences for what concerns our theoretical understanding of the out-of-equilibrium phenomena both in quantum and classical realms. It is worth explaining why.

    **The main idea.** We take advantage of the interplay between classical and quantum mechanics to further deepen our understanding in both realms. Indeed, as we extensively discuss below, the quantum world has been thoroughly investigated and several important results have been achieved: surprisingly, in several instances tackling directly the quantum system revealed to be simpler than the classical case. To this hand, quantum results can be carried to the classical realm through proper semiclassical limits, achieving new insights. On the other hand, many unsolved questions in the quantum case beg to be addressed: one among all is the determination of the steady-state after a quantum quench, which still lacks a general method for generic initial states (see Ref. [19] and references therein). As we will show, this problem can be instead efficiently semi-analytically solved in the classical case, providing a complete

solution of the quantum quench problem, within the semiclassical approximation.

Classical non-relativistic integrable models can be accessed thanks to the attractive features of quantum relativistic integrable field theories (see, for instance [20] and references therein), in particular the possibility to obtain for these theories exact information on their elastic and factorized $S$-matrix amplitudes [21–23], the spectrum of their excitations, the thermodynamics [24,25], the exact expressions of the matrix elements of local operators [26–28], together with their correlation functions at zero temperature [29, 30] and at finite temperature [31–34]. Since in addition to genuine coupling constants, any quantum field theory has inherently two important parameters – the speed of light $c$ and the Planck constant $\hbar$ – it should be possible to use the solvability of the integrable quantum field theories to get exact results for all those models which emerge by playing with these two parameters. This will be extensively used hereafter.

**Three integrable models.** It is useful to introduce the three integrable models which will accompany us in the rest of the paper.

- The first is the relativistic integrable quantum field theory, called the sinh-Gordon (ShG) model [22, 25, 28], with Hamiltonian given by

$$H_{\text{ShG}} = \int \mathrm{d}x \left[ \frac{1}{2}\Pi^2 + \frac{1}{2}(\partial_x\Phi)^2 + \frac{m_0^2 c^2}{g^2\hbar^2}(\cosh(g\Phi)-1) \right] \ , \quad \Pi(t,x) \equiv c^{-1}\partial_t\Phi, \quad (1)$$

  where $\Phi = \Phi(t,x)$ is a real quantum scalar field which satisfies the canonical equal-time commutation relations

$$[\Phi(t,x),\Pi(t,y)] = i\hbar\,\delta(x-y) \ , \quad [\Phi(t,x),\Phi(t,y)] = 0 \ .$$

  Of course, replacing commutators with Poisson brackets, $[,] \to i\hbar\{,\}$, and regarding $\Phi(t,x)$ as a classical field, one has the classical ShG model.

- The second is the integrable non-relativistic quantum field theory, called the Lieb-Liniger (LL) model [35–37], with Hamiltonian given by

$$\hat{H} = \int \mathrm{d}x \left[ \frac{\hbar^2}{2m}\partial_x\psi^\dagger\partial_x\psi + \kappa\hbar^4\,\psi^\dagger\psi^\dagger\psi\psi \right], \quad (2)$$

  where the complex Bose field $\psi(t,x)$ satisfies the canonical commutation relations

$$[\psi(x),\psi^\dagger(y)] = \hbar\delta(x-y) \ , \quad [\psi(x),\psi(x')] = 0 \ .$$

  In the Hamiltonian (2), the interaction is rescaled in order to attain a meaningful semiclassical limit, as we will extensively discuss: while sending $\hbar \to 0$, $\kappa$ must be kept constant. In the following, we consider only the case $\kappa > 0$, i.e. the repulsive regime of the Lieb-Liniger model. The Fock vacuum $|0\rangle$ of this field theory is defined by $\psi(t,x)|0\rangle = 0$. Therefore $\psi(t,x)$ and $\psi^\dagger(t,x)$ are respectively annihilation and creation operators. The LL Hamiltonian describes the dynamics of $N$ bosonic particles interacting only through a very local potential, a situation which is quite common in cold atom systems. Indeed, the Lieb-Liniger model is a milestone in our understanding of experiments with cold atoms and is nowadays realised in several instances [3–18].

- The third and last model is the integrable non-relativistic and classical field theory, called Non-Linear Schrödinger (NLS) model [38, 39], with Hamiltonian given by

$$H = \int \mathrm{d}x \left[ \partial_x\phi^\dagger\partial_x\phi + \phi^\dagger\phi^\dagger\phi\phi \right], \quad (3)$$

and Poisson brackets $\{\phi(x), \phi^{\dagger}(y)\} = \iota \delta(x-y)$. This gives rise to the classical equation of motion

$$\iota \partial_t \phi = -\partial_x^2 \phi + 2\phi^{\dagger}\phi\phi\,. \tag{4}$$

The NLS equation is a prototypical dispersive nonlinear partial differential equation which, in addition to its application in cold-atom physics, enters the phenomenon of self-focusing and the conditions under which an electromagnetic beam can propagate without spreading in nonlinear media [40].

In writing down the Hamiltonian (3) we have used the freedom to change the normalization of the classical field $\phi(t, x)$ to absorb the coupling constant in front of the quartic term. It is worth noticing the close similarity between the two Hamiltonians (2) and (3). In the following we will use the notation $\psi(t, x)$ to denote the *quantum* non-relativistic field of the LL model while $\phi(t, x)$ to denote the *classical* non-relativistic field of the NLS model. In the same spirit, we denote the quantum Hamiltonian (2) with the "hat", since it deals with quantum objects, while we remove the "hat" for the NLS Hamiltonian (3).

**Playing with $c$ and $\hbar$.** Let's now use the three models above to illustrate our previous observation, their connections is depicted in Fig. 1. For instance, taking $c \to \infty$ of the integrable sinh-Gordon model, one can address the exact computation of local quantities of the Lieb-Liniger model, since the latter is the non-relativistic limit of the former [41, 42] (see also Refs. [43, 44]): this mapping between the two models allows one to explicitly get the analytic expression of one-point $n$-body correlation functions of the Lieb-Liniger model and many other quantities, among which, the full counting statistics for the particle-number fluctuations in a short interval [45–47]. Taking instead the $\hbar \to 0$ limit in a quantum field theory, it opens the way to study the (semi) classical theory both at equilibrium and out of equilibrium which emerges in this limit [48]. From a physical point of view, the (semi) classical behaviour of a quantum field theory controls the regime when the occupation numbers of the various modes of the field are very high, typically at very high temperatures. Handling the $\hbar \to 0$ limit has proved moreover to be particularly rich and intriguing in light of the fact that in this limit one can establish an interesting dictionary between the quantum and classical integrable worlds out of equilibrium. Beside the theoretical insights gained in this way on both subjects, this approach seems also to be more efficient and better suited for managing all those classical situations out of equilibrium in which the energy scales are macroscopic and extensive with the volume of the system: in these cases, in fact, there are an extensive number of excitations involved in the out of equilibrium dynamics and, in the traditional approach to the problem, this would require to employ the notoriously difficult infinite-gap solutions of the inverse scattering method [38, 39, 49].

As discussed in detail below, in this paper we are going to extend the above analysis and tackle the asymptotic steady state of the out of equilibrium dynamics of the Non-Linear Schrödinger model.

We look at this problem from two directions: on one hand, we use known facts in the Lieb-Liniger model and the $\hbar \to 0$ limit to achieve new exact results for the moments of the density field $|\phi|^2$ in the Non-Linear Schröedinger. It is worth saying that the mentioned results in the LL model have been obtained taking the non-relativistic limit of the quantum sinh-Gordon model [46, 47]. On the other hand, we use purely classical arguments on the NLS to determine the steady state after a quench in terms of the initial conditions.

This leads us, in particular, to the prediction of the asymptotic values of several observables as well as of the full counting statistics of the classical field. In order to appreciate better the general background in which this paper is placed and the nature of questions we are going to address, it is also useful to remind a few important aspects of out of equilibrium quantum physics.

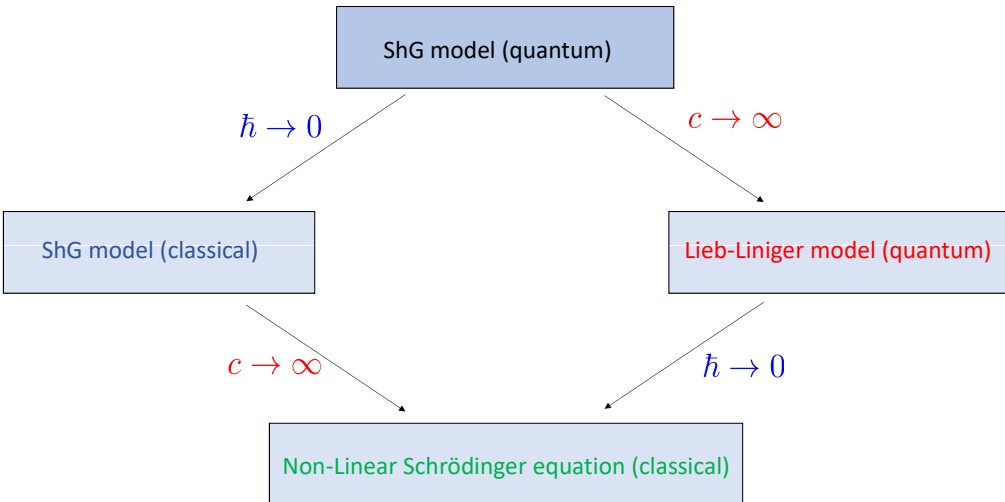

Figure 1: Mapping of the various theories related to the quantum field theory of the sinh-Gordon model.

**Properties of the steady state.** After a quantum quench, the quantum system undergoes a unitary time evolution. Yet, under quite general assumptions, one expects that the system will *locally* relax to a certain steady-state which can be characterised in terms of statistical physics. The corresponding ensemble depends though on the nature of the system. If the system under consideration has only its Hamiltonian $\hat{H}$ as an extensive integral of motion, it will locally equilibrate to the familiar Gibbs Ensemble (GE)

$$\hat{\rho}_{\mathrm{GE}} = \frac{1}{\mathcal{Z}} e^{-\beta \hat{H}}, \tag{5}$$

whose temperature is determined by the energy injected at $t = 0$, which is conserved during the time evolution. On the contrary, if the system is one of those interacting models which are integrable [20, 26, 37], i.e. those models which, besides the Hamiltonian itself, possess an extensive number of local (and quasi-local) integrals of motion $\hat{\mathcal{Q}}_j$ [50–58], the late-time steady state will be described by Generalised Gibbs Ensemble [59] (GGE)

$$\hat{\rho}_{\mathrm{GGE}} = \frac{1}{\mathcal{Z}} e^{-\sum_j \beta_j \hat{\mathcal{Q}}_j}, \tag{6}$$

where the infinitely many generalised temperatures $\beta_j$ are in principle fixed matching the expectation values of the charges $\hat{\mathcal{Q}}_j$ in terms of their value on the initial state. In this case, the presence of infinitely-many conserved charges will deeply constrain the dynamics of these models and many properties of the initial states are not washed out by the time-evolution, but remain encoded in the statistical nature of late-time steady state.

**GGE: *cahiers de doleances*.** It is important to remark that the actual determination of the $\beta_j$'s from the initial conditions is usually extremely difficult: besides free-to-free quenches (which can be exactly solved by means of Bogoliubov rotations), exact results can be obtained only for

very special cases (see Ref. [1] and references therein for a comprehensive overview), leaving out of the analysis most of the possible physically-motivated quenches. From this point of view, any efficient method able to control the GGE, no matter if approximated or based on numerical approaches, would be really welcome.

Among the methods used so far, one consists in approximating the GGE by employing only a finite number of charges: this is the idea behind the Truncated GGE (TGGE) [60] and the expectation is that, increasing the number of the charges in this truncation, the corresponding ensemble will converge to the true GGE. While such a method can be successfully implemented in some cases, several pitfalls can undermine its applicability. First of all, one should know the whole set of charges: even though many integrals of motion are often not too hard to be explicitly constructed, determining the whole set of relevant charges is a quest on its own. Indeed, failing to include all the relevant charges led to questioning the applicability of the GGE itself [61], which has been lately redeemed by the discovery of quasi-local charges [50, 52, 62, 63].

Another problem mining the TGGE approach is that the expectation value of the charges could be not-informative at all. A famous example in this context concerns the interacting Bose gas described by the Lieb-Liniger Hamiltonian (2). If one initializes the system in the non-interacting ground state and quenches to the interacting case ($\kappa = 0 \to \kappa \neq 0$), one immediately realises that the expectation values of all the local charges are indeed actually UV divergent [64] (albeit lattice regularizations can be attempted to overcome this issue [65–67]), obstructing the applicability of the TGGE. Let's mention that this drawback of the LL model has been lately explicitly overcome through the Quench Action Method [68, 69], which relies on the exact computation (in the thermodynamic limit) of the overlaps between the post-quench basis and the pre-quench state [70]. Lately, this method has been used in other models as well (see Ref. [19] and references therein), but its applicability seems to be confined to a restricted set of states [19, 71]) while more general (and physical) initial states are still waiting for a solution.

Numerical techniques [72] based on the Quench Action Method have been devised resulting in the ABACUS algorithm [73]: while being able to capture the whole time evolution for quenches with arbitrary interaction changes, its efficiency is drastically reduced for thermal initial states at sufficiently small temperature. In this case, higher temperatures are more difficult to access.

**Organization of the paper and main results.** In this paper, we are going to revisit the problem through a *semiclassical* approach, which well approximates the quantum model in the limit of high density and energy, as we thoroughly discuss later on. Semiclassical methods have been applied in a variety of contexts [74–80] and, for what concerns the Lieb-Liniger model, gave access to predictions unavailable in the purely quantum context. Following the original paper [48], the (semi)classical analysis of integrable field theories out of equilibirum has been extended to weakly-inhomogeneous integrable models [81, 82] in the context of the Generalised Hydrodynamics [83, 84].

In the following, we focus on the classical counterpart of the Lieb-Liniger model in the repulsive phase ($\kappa > 0$), namely the Non-Linear Schrödinger (NLS) equation, and consider quenches from arbitrary homogeneous initial states to the repulsive phase.

- In Sec. 2 we introduce the main character of our analysis, namely the LL model: we briefly review its thermodynamics and its semiclassical limit, i.e. the NLS equation.

- In order to study out-of-equilibrium setups, in Sec. 3 we provide a new analytical toolbox to determine physical observables. More specifically, we determine new close integral expressions for arbitrary moments of the particles' density valid on arbitrary GGEs (see Refs. [46, 47] for the quantum result). Furthermore, we also analytically derive exact

expressions for the whole probability distribution of the density operator for any GGE, a quantity also known as Full Counting Statistics (FCS). The FCS of several quantities and in a variety of models have been investigated in several instances [85–100], but for what concerns the LL model only a few results have been obtained so far, despite their relevance in experiments [101–103]. For instance, in Ref. [46,47], the FCS for the number of particles in a small interval has been computed in the fully quantum Lieb-Liniger model, at the first order in the size of the interval. On the contrary, in Ref. [75] the FCS for the number of particles for arbitrary intervals has been considered within the semiclassical approximation, but the validity of the result is restricted to thermal ensembles.

- Sec. 4 is devoted to determining the root density from the initial conditions of the out-of-equilibrium protocols. We start presenting the integrability of the NLS from a purely classical perspective and then, building on the findings of Ref. [48], we propose an efficient numerical algorithm for the determination of the GGE. More precisely, the root density is numerically determined evaluating a particular observable, called transfer matrix, which is extensively discussed in this section.

- Sec. 5 is devoted to benchmarking our predictions with first-principle numerical simulations, while in Section 6 we gather our conclusions. A few appendixes follow the main text, providing a few more technical discussions on several aspects touched in this paper.

## 2 The Lieb-Liniger model

In this Section, we present some basic facts about the Lieb-Liniger model, in particular we characterise the Hilbert space and the thermodynamics of this model, as well as the expectation values of some physical observables. For the computation of the latter quantities it is worth underlying that the LL model emerges as the non-relativistic limit of the sinh-Gordon model [41,42] or, more generally, of the Toda Field Theories [43]. Given the integrability of the model (see, for instance Ref. [37]), one can look for the common eigenvectors of the whole set of conserved charges. As shown in the original papers by Lieb and Liniger [35,36], these common eigenvectors consist of asymptotically free multi-particle states and their explicit expressions can be found by means of the Coordinate Bethe Ansatz.

**Eigenfunctions and eigenvalues.** The eigenfunctions can be written as

$$|\{\lambda_j\}_{j=1}^N\rangle = \int d^N x \, \chi_N(x_1, ..., x_N) \psi^\dagger(x_1)...\psi^\dagger(x_N)|0\rangle \,, \tag{7}$$

where the rapidities $\{\lambda_j\}_{j=1}^N$ parametrise the many-body wave-function

$$\chi_N(x_1, \ldots, x_N) = \sum_P A(P) \prod_{j=1}^N e^{i\lambda_{P_j} x_j} \,, \qquad x_1 \le x_2 \le ... \le x_N \,. \tag{8}$$

Above, the sum is over all the permutations $P$ of the rapidities and $\chi$ is symmetrically extended when the coordinates are reshuffled in a different order. The coefficients $A(P)$ bear the information about the non-trivial scattering: let $\Pi_{j,j+1}$ be the permutation swapping the rapidities at positions $j$ and $j + 1$, one has

$$A(\Pi_{j,j+1}P) = S(\lambda_{P_j} - \lambda_{P_{j+1}})A(P) \qquad , \qquad S(\lambda) = \frac{\lambda + i2m\hbar^2\kappa}{\lambda - i2m\hbar^2\kappa} \,, \tag{9}$$

with $S(\lambda)$ the Lieb-Liniger two-body scattering matrix. Eq. (9) makes clear the factorization of the multi-particle scatterings in terms of two-body scattering amplitudes. As we already mentioned, the states (7) are common eigenvectors of all the conserved charges. Besides, the latter act additively on the rapidities, enforcing once again the multi-particle interpretation. For example, for what concerns the energy and the number of particles $\hat{N} = \int dx\, \psi^\dagger \psi$ one gets

$$\hat{H}|\{\lambda_j\}_{j=1}^N\rangle = \left(\sum_{j=1}^N E(\lambda_j)\right)|\{\lambda_j\}_{j=1}^N\rangle \qquad , \qquad \hat{N}|\{\lambda_j\}_{j=1}^N\rangle = N|\{\lambda_j\}_{j=1}^N\rangle, \qquad (10)$$

where $E(\lambda) = \frac{\hbar^2}{2m}\lambda^2$ is the single-particle energy. In the case of other charges, one gets similar expressions replacing the energy with the proper single-particle eigenvalue of the charge.

**Thermodynamics.** In order to study the thermodynamic limit of the model, it is crucial to consider a finite system of length $L$ and then take the infinite volume limit, while consequentially rescaling the number of particles in such a way the density $n = N/L$ remains fixed. Hence, we consider the bosons to live on a ring with periodic boundary conditions (PBC): different choices of boundary conditions do not affect the physics in the thermodynamic limit. Analogously to the non-interacting case, where putting the system in a finite volume results in a quantization of the allowed wavevectors, in the interacting case rapidities are constrained by the Bethe-Gaudin equations

$$e^{i\lambda_j L}\prod_{k\neq j}^N S(\lambda_j - \lambda_k) = 1. \qquad (11)$$

Apart from the limiting case of free ($\kappa = 0$) or hard-core ($\kappa \to +\infty$) bosons, the rapidities are coupled together in a highly non-linear manner and an analytical solution of these equations for large $N$ is hopeless. Fortunately, in the thermodynamic limit we can leave aside the attempt of any exact solution and adopt a coarse-grained approach, which is handled by the Thermodynamic Bethe Ansatz [104] (TBA) method. In the repulsive regime of the LL model, the relevant solutions of the Bethe-Gaudin equations (11) have all real rapidities and, rather than exactly tracking them, we introduce a coarse-grain counting functions $\rho_q(\lambda)$, called the *root density*. Since later on we will extensively discuss the classical counterpart of the LL model (namely the NLS equation), we find it useful to add a label "q" to denote quantities in the quantum case while, without labels, we will refer to the classical case.

Given a physical state as in Eq. (7), $L d\lambda \rho_q(\lambda)$ counts the number of rapidities laying around $\lambda$ in a small window of length $d\lambda$. The root density fully encodes the extensive expectation value of the charges: for what concerns the energy and number of particles, one has

$$L^{-1}\langle\{\lambda_j\}_{j=1}^N|\hat{H}|\{\lambda_j\}_{j=1}^N\rangle = \int d\lambda\, E(\lambda)\rho_q(\lambda) + \mathcal{O}(L^{-1}) \qquad (12a)$$

$$L^{-1}\langle\{\lambda_j\}_{j=1}^N|\hat{N}|\{\lambda_j\}_{j=1}^N\rangle = \int d\lambda\, \rho_q(\lambda) + \mathcal{O}(L^{-1}), \qquad (12b)$$

and similarly for the other charges. It is however necessary to introduce another quantity, $\rho_q^t(\lambda)$, also known as *total density*, which refers to the total number of available modes in the interval $(\lambda, \lambda + d\lambda)$, while $\rho_q(\lambda)$ counts only the occupied modes which are solutions of (11) and appear in the multi-particle state (7). These two densities are related to each other through the integral equation (11)

$$\rho_q^t(\lambda) = \frac{1}{2\pi} + \int \frac{d\lambda'}{2\pi}\varphi(\lambda - \lambda')\rho_q(\lambda') \ , \qquad \varphi(\lambda) = \frac{4m\hbar^2\kappa}{\lambda^2 + (2m\hbar^2\kappa)^2}. \qquad (13)$$

Together with the root density, it is customary to define the filling

$$\vartheta_q(\lambda) = \frac{\rho_q(\lambda)}{\rho_q^t(\lambda)} = \frac{1}{e^{\varepsilon_q(\lambda)} + 1} \ , \tag{14}$$

and parameterizing it in terms of the effective energy $\varepsilon_q(\lambda)$. In the particular case of a thermal ensemble with inverse temperature $\beta_q$ and chemical potential $\mu$, the effective energy satisfies the non-linear integral equation [104]

$$\varepsilon_q(\lambda) = \beta_q \big[ E(\lambda) - \mu \big] - \int_{-\infty}^{\infty} \frac{d\lambda'}{2\pi} \varphi(\lambda - \lambda') \log \big( 1 + e^{-\varepsilon_q(\lambda')} \big) \ . \tag{15}$$

**The importance of the root density.** The TBA can be easily generalised to GGEs by including higher charges, provided the whole set of generalised temperatures $\beta_j$ is known [105, 106]. Notice that, given a GGE, the root density is fully specified, but the implication holds true also the other way around. Indeed, in Ref. [52] it has been shown that the set of possible GGEs is in one-to-one correspondence with the root densities: in this respect, determining the GGE in the form Eq. (6) is completely equivalent to find the correct root density. This second point of view must be preferred to the previous one, since it does not require to know all the charges explicitly: hereafter, we will always refer to GGEs just specifying the root density (or equivalently the filling). Once the root density is known, one can easily compute the expectation value of the charges (12), but the local relaxation to a GGE implies much more: any local property of the system, such as the expectation values of local observables and correlation functions thereof, is fully specified by the GGE or, equivalently, by the root density.

**Moments of the density operator.** While the expectation values of the charges are rather simple, determining other observables (which, in contrast with the charges, undergo a non trivial dynamics after the quench) is in general a hard task. In Refs. [46, 47] the problem has been solved for all the moments of the density operator, i.e. $\langle (\psi^\dagger(x))^n (\psi(x))^n \rangle$, for arbitrary positive integers $n$ and the expectation value can be taken on an arbitrary GGE (see also Ref. [45, 107] for previous results up to $n = 4$). Since this result will play a key role in the forthcoming discussion of the classical model, it is worth to report it shortly. The expectation values can be obtained expanding in the dummy variable $Y$ the following generating function (we slightly change the notation compared with the original references Refs. [46, 47], also in view of the forthcoming semiclassical limits )

$$1 + \sum_n Y^n \frac{2^n (2m\hbar^2\kappa)^n}{(n!)^2} \langle (\psi^\dagger)^n (\psi)^n \rangle = \exp \left( \frac{1}{\pi} \sum_{n=1} Y^n \mathcal{G}_n^q \right) \tag{16a}$$

$$\mathcal{G}_n^q = \frac{(2m\hbar^2\kappa)^{2n-1}}{n} \int d\lambda \, \vartheta_q(\lambda) \xi_{2n-1}^{dr(q)}(\lambda), \tag{16b}$$

where the auxiliary functions $\xi_n(\lambda)$ satisfy the following recursive set of linear integral equations (below, we define $\Gamma(\lambda) = \lambda (2m\hbar^2\kappa)^{-1} \varphi(\lambda)$)

$$\xi_{2n}(\lambda) = \int \frac{d\lambda'}{2\pi} \vartheta_q(\lambda') \Big\{ \Gamma(\lambda - \lambda') [2\xi_{2n-1}^{dr(q)}(\lambda') - \xi_{2n-3}^{dr(q)}(\lambda')] - \varphi(\lambda - \lambda') \xi_{2n-2}^{dr(q)}(\lambda') \Big\} \tag{17}$$

$$\xi_{2n+1}(\lambda) = \delta_{n,0} + \int \frac{d\lambda'}{2\pi} \vartheta_q(\lambda') \Big\{ \Gamma(\lambda - \lambda') \xi_{2n}^{dr(q)}(\lambda') - \varphi(\lambda - \lambda') \xi_{2n-1}^{dr(q)}(\lambda') \Big\} . \tag{18}$$

Above, for an arbitrary test function $\omega(\lambda)$, we define the (quantum) dressing operation as

$$\omega^{dr(q)}(\lambda) = \omega(\lambda) + \int \frac{d\lambda'}{2\pi} \varphi(\lambda - \lambda') \omega^{dr(q)}(\lambda') \vartheta_q(\lambda') . \tag{19}$$

These equations are iteratively solved posing $\xi_{n \le 0} = 0$ and increasing $n$ step by step. The whole information about the GGE is contained in the filling function $\vartheta_q$.

# 3 The NLS as the semiclassical limit of the LL

Let us now explore the connection between the Lieb-Liniger model and its classical counterpart, namely the Non Linear Schroedinger equation. We denote with $\phi(t, x)$ the classical complex field which obeys the equation of motion (4). As a warm up to understand the applicability of the semiclassical approximation, it is useful to look at thermal ensembles of the LL model without referring to its integrability structure. As we have already commented in the introduction, the semiclassical limit can be viewed as a high temperature limit. In order to see this, we define a rescaled Hamiltonian

$$\hat{H}' = \int dx \left( \partial_x \psi^\dagger \partial_x \psi + 2m\kappa\hbar^2 \psi^\dagger \psi^\dagger \psi\psi \right). \tag{20}$$

Comparing with the Hamiltonian (2), we have $\hat{H} = \frac{\hbar^2}{2m}\hat{H}'$, hence the equality of the density matrices

$$e^{-\beta_q \hat{H}} = e^{-\beta_q' \hat{H}'}, \tag{21}$$

with

$$\beta_q' = \frac{\hbar^2}{2m}\beta_q. \tag{22}$$

As it is clear, the $\hbar \to 0$ can be interpreted as a high temperature/small coupling limit of for the rescaled Hamiltonian: we will pursue this second interpretation along this section.

**Partition functions.** Let us initially consider the (quantum) partition function for the LL model. We consider the rescaled Hamiltonian (20) with an inverse temperature $\beta_q'$ (22) and a chemical potential $\mu$, describing the partition function within a path integral approach. There is a standard procedure to reach the semiclassical limit of this expression (see, for instance [108]) and this consists of supplementing the real space with a Euclidean time coordinate $\tau$. Assuming periodic boundary conditions in the real space, the quantum problem is mapped into a classical partition function on a torus $(x, \tau) \in [0, L] \times [0, \beta_q']$ (see Fig. 2): for historical reasons, we refer to this domain as the Matsubara torus. The quantum partition function can then be written as

$$\mathcal{Z}_q = \int \mathcal{D}\psi \exp\left[ -\int_0^L dx \int_0^{\beta_q'} d\tau \left\{ \frac{1}{2}(\psi^\dagger \partial_\tau \psi - \psi \partial_\tau \psi^\dagger) + \partial_x \psi^\dagger \partial_x \psi \right.\right. \tag{23}$$
$$\left.\left. -\mu\psi^\dagger\psi + 2m\hbar^2\kappa\psi^\dagger\psi^\dagger\psi\psi \right\} \right],$$

where $\psi(\tau, x)$ now denotes a classical field whose domain is the Matsubara torus. On the other hand, the classical partition function can be written as a path integral on a ring $x \in [0, L]$ (see Fig. 2)

$$\mathcal{Z} = \int \mathcal{D}\phi \exp\left[ -\beta \int_0^L dx \left\{ \partial_x \phi^\dagger \partial_x \phi - \mu\phi^\dagger\phi + \phi^\dagger\phi^\dagger\phi\phi \right\} \right]. \tag{24}$$

In order to map the quantum partition function into a classical one, we should be able to "shrink" the imaginary time dimension $\beta_q' \to 0$ (see again Fig. 2): this is indeed what the $\hbar \to 0$ limit does, as it is clear from Eq. (22). To do so, let us consider the Fourier components of the field along the Euclidean time direction

$$\psi(\tau, x) = \sum_{n \in \mathbb{Z}} \frac{e^{i2\pi n\tau/\beta_q'}}{\sqrt{2m\hbar^2\kappa}} \psi_n(x). \tag{25}$$

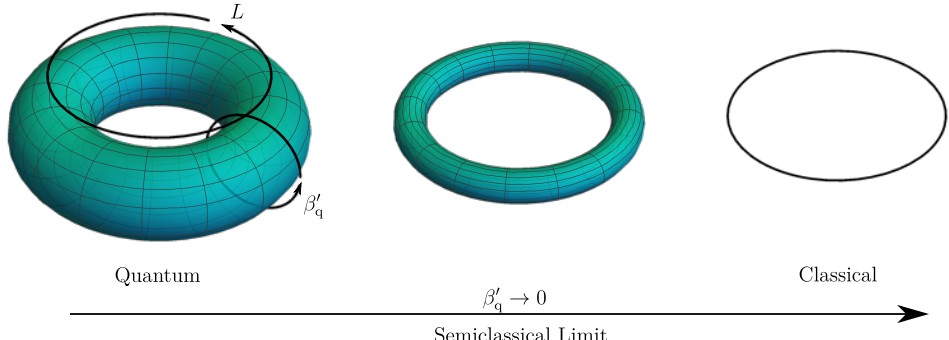

Figure 2: Graphical representation of the semiclassical limit of the thermal partition function. Within the path integral formalism, the partition function of the *quantum* Lieb-Liniger model can be represented as a classical partition function on a two-dimensional torus $[0, L] \times [0, \beta'_q]$. Increasing the temperature, i.e. $\beta'_q \to 0$, the transverse dimension of the torus shrinks until the torus collapses on a one dimensional ring, namely the domain of the path integral describing the *classical* partition function of the Non-Linear Schröedinger equation.

Above, for later use, we have suitably normalised the modes: this will introduce an overall constant renormalization of the partition function which however does not affect the thermodynamics. This is dominated by the zero-frequency mode and therefore it is described by the classical NLS. Indeed, plugging this mode decomposition in the quantum action, one finds

$$\mathcal{Z}_q = \int \mathcal{D}\psi \exp\left[ -\frac{\beta'_q}{2m\hbar^2\kappa} \int_0^L \mathrm{d}x \sum_n \left\{ (i2\pi/\beta'_q)n\psi_n^\dagger\psi_n + \partial_x\psi_n^\dagger\partial_x\psi_n - \mu\psi_n^\dagger\psi_n \right\} \right.$$
$$\left. + \sum_{n_1+n_2=n_3+n_4} \psi_{n_1}^\dagger\psi_{n_2}^\dagger\psi_{n_3}\psi_{n_4} \right] \tag{26}$$

and, when $\beta'_q \to 0$, this path integral is dominated by the zero-frequency Matsubara mode since the $(i2\pi/\beta'_q)n\psi_n^\dagger\psi_n$ term tends to pin $\psi_{n\neq 0} \to 0$, while it vanishes for $\psi_0$ that is then free to fluctuate. Hence, discarding the non-vanishing Matsubara frequencies we can approximate the partition function as

$$\mathcal{Z}_q \simeq \int \mathcal{D}\psi_0 \exp\left[ -\frac{\beta'_q}{2m\hbar^2\kappa} \int \mathrm{d}x \left\{ \partial_x\psi_0^\dagger\partial_x\psi_0 - \mu\psi_0^\dagger\psi_0 + \psi_0^\dagger\psi_0^\dagger\psi_0\psi_0 \right\} \right], \tag{27}$$

which is nothing else than the classical partition function Eq. (24), provided the replacement

$$\phi(x) \to \psi_0(x), \qquad \beta \to \frac{\beta'_q}{2m\hbar^2\kappa} = \frac{\beta_q}{(2m)^2\kappa}. \tag{28}$$

Notice that the classical inverse temperature $\beta$ is kept constant in the $\hbar \to 0$ limit, as it should be.

**TBA in the semiclassical regime.** After this warm up on the partition functions, let's now move on to discuss the semiclassical limit from the integrability point of view within the TBA framework, extending the original discussion done in Ref. [48] (see also Ref. [83]). We will now use the quantum TBA to access the classical one, together with other results. Once the

mapping between the quantum and the classical TBA has been established, in the following we will refer to the classical TBA simply as the TBA of the NLS model. As already stated, the classical physics emerges from the quantum one in the limit of high occupation numbers, which in free systems is the mode density: within the interacting integrable framework, the mode density is replaced by the root density, which is then expected to diverge with $\hbar$. Inspired by the thermal case and using Eq. (25), one immediately sees that in the semiclassical limit the density of particles diverges as $\propto (2m\kappa\hbar^2)^{-1}$: this same behavior is reproduced at the level of TBA if one poses

$$\rho_{\mathrm{q}}(\lambda) = (2m\kappa\hbar^2)^{-1}\rho(\lambda), \tag{29}$$

with $\rho(\lambda)$, interpreted as the classical root density, kept constant while sending $\hbar \to 0$. Let us use this scaling to extract the semiclassical limit of the basic building blocks of the TBA: we will be back to the thermal state later on. We start defining the classical total root density: in this respect we consider Eq. (13) with the replacement (29). As it stands, the left and right hand sides do not have a well defined $\hbar \to 0$ limit: using the identity $\rho(\lambda) = \int d\lambda' \, \delta(\lambda - \lambda')\rho(\lambda')$ we can recast Eq. (13) in the following equivalent form

$$\rho_{\mathrm{q}}^t(\lambda) - \frac{\rho(\lambda)}{2m\hbar^2\kappa} = \frac{1}{2\pi} + \int \frac{d\lambda'}{2\pi}\Big[\varphi(\lambda - \lambda') - 2\pi\delta(\lambda - \lambda')\Big](2m\hbar^2\kappa)^{-1}\rho(\lambda'). \tag{30}$$

Note that $\varphi(\lambda) - 2\pi\delta(\lambda) = \partial_\lambda\big[2\arctan\big(\lambda(2m\hbar^2\kappa)^{-1}\big) - \pi\,\mathrm{sign}(\lambda)\big]$. By means of a simple integration by parts, we obtain the following equivalent form, which is better suited for taking the semiclassical limit

$$\rho_{\mathrm{q}}^t(\lambda) - \frac{\rho(\lambda)}{2m\hbar^2\kappa} = \frac{1}{2\pi} + \int \frac{d\lambda'}{2\pi}\Big[2\arctan\Big(\frac{\lambda - \lambda'}{2m\hbar^2\kappa}\Big) - \pi\,\mathrm{sign}(\lambda - \lambda')\Big]\frac{\partial_{\lambda'}\rho(\lambda')}{2m\hbar^2\kappa}. \tag{31}$$

Taking the $\hbar \to 0$ limit one has

$$\lim_{\hbar \to 0} \int \frac{d\lambda'}{2\pi}\Big[2\arctan\Big(\frac{\lambda - \lambda'}{2m\hbar^2\kappa}\Big) - \pi\,\mathrm{sign}(\lambda - \lambda')\Big]\frac{\partial_{\lambda'}\rho(\lambda')}{2m\hbar^2\kappa} = -\fint \frac{d\lambda'}{2\pi}\frac{2}{\lambda - \lambda'}\partial_{\lambda'}\rho(\lambda'), \tag{32}$$

where the singular integral is handled with the principal part prescription. We can now define the classical total root density trough the limit $\rho^t(\lambda) = \lim_{\hbar \to 0}[\rho_{\mathrm{q}}^t(\lambda) - (2m\hbar^2\kappa)^{-1}\rho(\lambda)]$, leading to the integral equation

$$\rho^t(\lambda) = \frac{1}{2\pi} - \fint \frac{d\lambda'}{2\pi}\frac{2}{\lambda - \lambda'}\partial_{\lambda'}\rho(\lambda'), \tag{33}$$

which defines the total root density in the classical model. In analogy with the quantum case, one defines the classical filling $\vartheta(\lambda) = \rho(\lambda)/\rho^t(\lambda)$: this definition, together with Eq. (33), is consistent with the one already given in Ref. [48]. A similar analysis can be performed on the dressing (19) and on the equation for the thermal state (15): the classical dressing on a test function $\omega(\lambda)$ is defined as

$$\omega^{\mathrm{dr}}(\lambda) = \omega(\lambda) - \fint \frac{d\lambda'}{2\pi}\frac{2}{\lambda - \lambda'}\partial_{\lambda'}\big[\omega^{\mathrm{dr}}(\lambda')\vartheta(\lambda')\big] \tag{34}$$

and one has the following quantum-classical correspondence

$$\lim_{\hbar \to 0}\big[2m\hbar^2\kappa\,\vartheta_{\mathrm{q}}(\lambda)\omega^{\mathrm{dr(q)}}(\lambda)\big] = \vartheta(\lambda)\,\omega^{\mathrm{dr}}(\lambda). \tag{35}$$

Concerning the thermal states, one can still obtain convenient integral equations in terms of an effective energy $\varepsilon(\lambda)$, although its relation with the filling is modified with respect to the quantum case (14)

$$\vartheta(\lambda) = \frac{1}{\varepsilon(\lambda)}. \tag{36}$$

Then, the thermal state is determined by the following integral equation

$$\varepsilon(\lambda) = \beta\big[E(\lambda) - \mu\big] - \int \frac{\mathrm{d}\lambda'}{2\pi} \frac{2}{\lambda - \lambda'} \partial_{\lambda'} \log\big(\varepsilon(\lambda')\big), \qquad (37)$$

which can be readily derived from Eq. (15) in the $\hbar \to 0$ limit, as done in Eq. (28). The comparison between the expression of the classical filling Eq. (36) and the quantum one Eq. (14) in terms of the respective effective energies deserves further comments. Indeed, Eq. (36) is the generalization to the interacting integrable case of the Rayleigh-Jeans distribution, which is obeyed by the classical fields at equilibrium and leads to the famous UV catastrophe, namely to an UV divergence of the energy expectation value $\langle H \rangle = \int \mathrm{d}\lambda\, E(\lambda)\rho(\lambda)$. Indeed, for large rapidities, Eq. (37) gives $\varepsilon(\lambda) \sim \beta E(\lambda)$ and, in view of Eq. (36), one readily gets $\rho(\lambda) \sim (\beta E(\lambda))^{-1}$ with a consequent UV divergence of the energy. The UV catastrophe is what led Planck to discover quantum mechanics: in the quantum case Eq. (14) the Fermi-Dirac distribution gives a finite energy and regularizes the classical prediction.

The UV divergence of the local charges on a physical state, such as a thermal ensemble, suggests that they are not the best observables to look at. Furthermore, being our main interest out-of-equilibrium protocols, we wish to study observables which have a non-trivial time evolution after the quench: in this way, we probe the time evolution and observe the relaxation to the GGE. In view of all these considerations, we focus our attention on the moments of the density $|\phi(x)|^{2n}$: we now derive close integral expressions valid on arbitrary GGEs from the semiclassical limit of the analogue expression in the LL model [46, 47]. This is a new result in the context of the Non-Linear Schrödinger equation.

**The density moments from the semiclassical limit.** The semiclassical limit of the generating function Eq. (16) is straightforward. Indeed, from the analysis of the thermal case we know

$$\lim_{\hbar \to 0} (2m\hbar^2\kappa)^n \langle (\hat{\psi}^\dagger)^n (\hat{\psi})^n \rangle = \langle |\phi|^{2n} \rangle, \qquad (38)$$

which immediately leads to the classical generating function

$$1 + \sum_n Y^n \frac{2^n}{(n!)^2} \langle |\phi|^{2n} \rangle = \exp\left( \frac{1}{\pi} \sum_{n=1} Y^n \mathcal{G}_n \right), \qquad \mathcal{G}_n \equiv \lim_{\hbar \to 0} \mathcal{G}_n^{\mathrm{q}} = \frac{1}{n} \int \mathrm{d}\lambda\, \vartheta(\lambda)\, \zeta_{2n-1}^{\mathrm{dr}}(\lambda). \qquad (39)$$

Above, we plugged the correspondence Eq. (35) in the definition of $\mathcal{G}_n^{\mathrm{q}}$ (16) and defined

$$\zeta_n(\lambda) \equiv \lim_{\hbar \to 0} (2m\hbar^2\kappa)^{n-1} \xi_n(\lambda). \qquad (40)$$

As the last step, we plug the above definition of $\zeta_n$ in the integral equations for $\xi_n$ Eqs. (17) and (18) and take the semiclassical limit, resulting in the desired equations for the classical case

$$\zeta_n(\lambda) = \delta_{n,1} + \frac{3 + (-1)^n}{2} \int \frac{\mathrm{d}\lambda'}{2\pi} \frac{2}{\lambda - \lambda'} \vartheta(\lambda') \zeta_{n-1}^{\mathrm{dr}}(\lambda'). \qquad (41)$$

Within the quantum case, the expression for the density moments of Ref. [46, 47] has been found to be in agreement with previous results known in the literature up to the fourth moment [45, 107]. However, continuous quantum models are notoriously difficult to be simulated and these results lacked any numerical benchmark: the situation is different in the classical case, where the thermal expectation values can be efficiently computed by means of the Transfer Matrix (TM) method. We leave to Appendix A the description of the TM approach, while in Fig. 3 we benchmark the TBA results for different choices of temperature and chemical potential. Lately, in Section 5, we will use the density moments as a diagnostic tool to compare GGE's

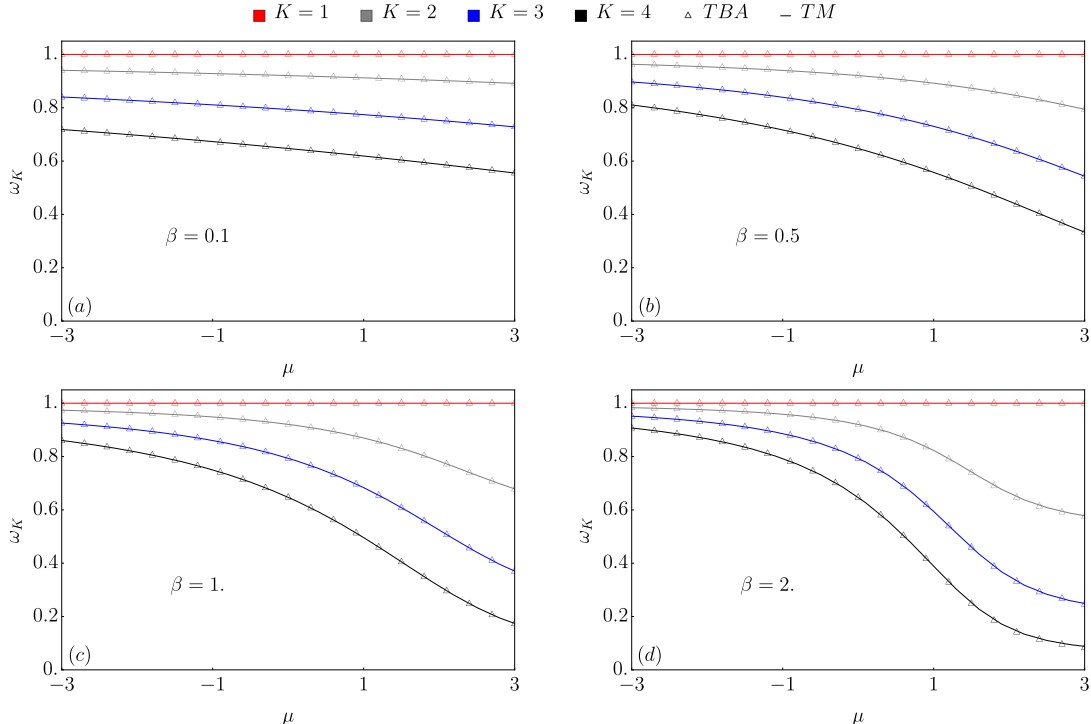

Figure 3: Comparison between the one-point functions $\omega_K = \langle|\phi|^{2K}\rangle/[(\langle|\phi|^2\rangle)^K K!]$ on thermal states as functions of the chemical potential for different inverse temperatures $\beta$ computed solving the TBA equations (37) (symbols) and using the Transfer Matrix approach (lines) (see Appendix A). The choice in the parametrization of the one-point observables is dictated by the free case: within the non-interacting case, the gaussianity of the thermal ensemble imposes $\omega_K = 1$, hence any departure from unity probes the effect of the interactions. Indeed, we notice that for smaller values of $\mu$ the density is reduced, thus the NLS approaches the non-interacting regime and $\omega_K \to 1$, as it should be. On the other hand, increasing $\mu$ the density grows, with the consequent enhancement of the interactions and $\omega_K$ significantly departs from unity.

computations with real-time simulations of quenches in the Non Linear Schroedinger equation, providing at the same time a check of the analytical results in other GGEs besides the thermal ensemble.

**The Full Counting Statistics.** Apart from the moments of the density $\langle|\phi(x)|^{2n}\rangle$, one could ask what is the probability to measure a certain value of $|\phi(x)|^2$. This is especially important when one can access a limited number of samplings or it is interested in the extreme-value statistics. Let us introduce $P(d)$ as the probability, on a given GGE, that a measurement gives $|\phi(x)|^2 = d$ as an output: formally, we can express it as the average of a Dirac$-\delta$ as follows

$$P(d) = \langle\delta(|\phi(x)|^2 - d)\rangle = \int \frac{\mathrm{d}\gamma}{2\pi} e^{-i\gamma d}\langle e^{i\gamma|\phi|^2}\rangle. \tag{42}$$

Above, the expectation value is taken with respect to a given GGE and the Fourier representation is introduced for later use. In the literature, the probability $P(d)$ is often called Full Counting Statistics (FCS) (in this case, FCS of the density operator) and it is notoriously hard to be analytically accessed. Here, we present an *exact* result for the classical FCS of the density operator valid for arbitrary GGEs. Therefore, at the price of leaving aside finite intervals, we

can generalise the result of Ref. [75] to out-of-equilibrium situations. In this case, an analogue result in the quantum case from which we can extract the semiclassical limit is not available and one must proceed from scratch. The starting point is an expression similar to Eq. (39), but of wider generality

$$\sum_{n=0}^{\infty} q^{2n} \frac{(16)^n}{(n!)^2} \langle |\phi|^{2n} \rangle = \exp\left[ \frac{16}{\pi} \int_0^q \mathrm{d}p\, p \int \mathrm{d}\lambda\, \vartheta(\lambda) W_p(\lambda) \right],$$ (43)

where

$$W_q(\lambda) = 1 + \int \frac{\mathrm{d}\lambda'}{2\pi} \frac{2}{\lambda - \lambda'} (4q - \partial_{\lambda'}) [\vartheta(\lambda') W_q(\lambda')].$$ (44)

In principle, one could recover Eq. (39) power-expanding in $q$ the r.h.s. of Eq. (43) (and the integral equation for $W_q$ as well). However, the other way around is far from being trivial: we derive Eq. (43) from known results in the classical sinh-Gordon model and taking the non relativistic limit. This derivation is rather technical and it is provided in Appendix B, while here we build on Eq. (43) in order to access the FCS. The analysis presented in Appendix B resulting in Eq. (43) assumes real parameters $q$, however in order to access the FCS we need to perform a proper analytical continuation. Let us introduce an auxiliary function $F(p)$ which has the following power expansion

$$F(p) = \sum_{n=0}^{\infty} p^n \frac{(16)^n}{(n!)^2} \langle |\phi|^{2n} \rangle.$$ (45)

We note that this is the same power expansion of Eq. (43) if one replaces $q^2 \to p$. Using the function $F(p)$, we can reach a convenient expression for the FCS. Indeed, from Eq. (45) one readily has

$$\partial_p^n F(p) \Big|_{p=0} = \frac{(16)^n}{n!} \langle |\phi|^{2n} \rangle$$ (46)

and establishes the following chain of equalities

$$\langle e^{i(\gamma+i0^+)|\phi|^2} \rangle = \sum_{n=0}^{\infty} \frac{[i(\gamma+i0^+)]^n}{n!} \langle |\phi|^{2n} \rangle = \sum_{n=0}^{\infty} \left[ \frac{i(\gamma+i0^+)}{16} \right]^n \partial_p^n F(p) \Big|_{p=0} =$$
$$\int \frac{\mathrm{d}k \mathrm{d}p'}{2\pi} \sum_{n=0}^{\infty} \left[ \frac{-k(\gamma+i0^+)}{16} \right]^n e^{-ikp'} F(p') = \int \frac{\mathrm{d}k \mathrm{d}p'}{2\pi} \frac{e^{-ikp'}}{1 + \frac{k(\gamma+i0^+)}{16}} F(p').$$ (47)

Above, we regularize the expression with an infinitesimal shift in the complex plane $\gamma \to \gamma + i0^+$. Using this result in the Fourier representation of the full counting statistics (42) and performing the integrals in $k$ and $\gamma$, one reaches the following representation of the FCS

$$P(d) = 16 \int_0^{\infty} \mathrm{d}\theta\, J_0(8\sqrt{\theta d}) F(-\theta).$$ (48)

Above, $J_0(x)$ is a modified Bessel function of the first kind. As it is clear from the above, the knowledge of the $F(p)$ function gives access to the FCS by means of a simple convolution. Going back to Eq. (43), one sees that the expression as it stands gives access to $F(p > 0)$ (as it is readily seen from Eq. (45)). However, extracting the FCS requires the knowledge of $F(p < 0)$: we fill this gap performing a proper analytical continuation of the r.h.s. of Eq. (45). While doing so, some subtleties arise. Let us proceed naively and promote $q$ to be imaginary in Eq. (43) posing $q = i\tau$, furthermore we define the auxiliary function $s_\tau(\lambda) = \vartheta(\lambda) W_{i\tau}(\lambda)$: by means of a direct replacement in Eq. (45) one finds

$$F(-\tau^2) \stackrel{?}{=} \exp\left[ -\frac{16}{\pi} \int_0^\tau \mathrm{d}\tau'\, \tau' \int \mathrm{d}\lambda\, s_{\tau'}(\lambda) \right],$$ (49)

where $s_\tau(\lambda)$ satisfies

$$\vartheta^{-1}(\lambda)s_\tau(\lambda) = 1 + \fint \frac{d\lambda'}{2\pi} \frac{2}{\lambda - \lambda'}(4i\tau - \partial_{\lambda'})s_\tau(\lambda'). \tag{50}$$

From the left hand side of Eq. (49), we see that the same result on the r.h.s. must be obtained if one replaces $\tau \to -\tau$. By a direct inspection of Eq. (50) we readily see $s_{-\tau}(\lambda) = s_\tau^*(\lambda)$, therefore after the replacement $\tau \to -\tau$ in Eq. (49) we get

$$F(-\tau^2) \stackrel{?}{=} \exp\left[-\frac{16}{\pi} \int_0^\tau d\tau' \tau' \int d\lambda\, s_{\tau'}^*(\lambda)\right]. \tag{51}$$

Imposing the consistency of the above with Eq. (49), we can conclude that $F(-\tau^2)$ must be real (which was expected, since the FCS must be a real function) and we must necessarily have

$$\int d\lambda\, s_\tau(\lambda) = \int d\lambda\, s_\tau^*(\lambda). \tag{52}$$

This equality can be easily shown as follows. We use a vector-matrix notation introducing a basis of states $|\lambda\rangle$ and defining an operator $\Omega(\tau)$ with matrix elements

$$\langle\lambda|\Omega(\tau)|\lambda'\rangle = \vartheta^{-1}(\lambda)\delta(\lambda - \lambda') - \frac{2}{\lambda - \lambda'}(4i\tau - i\partial_{\lambda'}). \tag{53}$$

We then introduce the state $|s_\tau\rangle$ such that $s_\tau(\lambda) = \langle\lambda|s_\tau\rangle$ and analogously the state $|1\rangle$ such that $\langle\lambda|1\rangle = 1$: matrix-vector contractions and scalar products are computed integrating over the $|\lambda\rangle$ states. In this notation Eq. (50) can be written as

$$|s_\tau\rangle = \Omega^{-1}(\tau)|1\rangle = \sum_n \frac{1}{\mu_n(\tau)}|n, \tau\rangle\langle n, \tau|1\rangle. \tag{54}$$

Above, we used the hermicity of $\Omega(\tau)$ and introduced a complete basis of eigenvectors $\Omega(\tau)|n_\tau\rangle = \mu_n(\tau)|n, \tau\rangle$. In the vector-matrix notation we have

$$\int d\lambda\, s_\tau(\lambda) = \langle 1|s_\tau\rangle = \sum_n \frac{1}{\mu_n(\tau)}|\langle n, \tau|1\rangle|^2, \tag{55}$$

which is clearly real, since $\Omega(\tau)$ has real eigenvalues. This proves Eq. (52). There is another subtlety in the analytic continuation, as it can be seen from Eq. (55): if one of the eigenvalues of the operator $\Omega(\tau)$ vanishes, the overlap $\langle 1|s_\tau\rangle$ develops a first order pole singularity as a function of $\tau$. Strictly speaking, such a singularity is not integrable and causes Eq. (49) to be ill-defined. This singularity arises in the analytical continuation while $q = i\tau$ approaches the imaginary axis: if we move slightly off from the imaginary axis the operator $\Omega(\tau)$ (53) is no longer real and, for infinitesimal shifts, the eigenvalues $\mu_n$ acquire a small imaginary part. This suggests to regularize Eq. (55) replacing $\mu_n(\tau) \to \mu_n(\tau) \pm i0^+$, both of sign choices leading to the same result, as we discuss below. Indeed, let us use the (regularized) expression (55) into Eq. (49), which results in

$$F(-\tau^2) \stackrel{?}{=} \exp\left[-\frac{16}{\pi} \int_0^\tau d\tau' \sum_n \frac{\tau'|\langle n, \tau'|1\rangle|^2}{\mu_n(\tau') \pm i0^+}\right] = \exp\left[-\frac{16}{\pi} \int_0^\tau d\tau' \left\{\left(\sum_n \frac{\tau'|\langle n, \tau'|1\rangle|^2}{\mu_n(\tau')}\right)\right.\right.$$
$$\left.\left. - \sum_{j, \tau_j \le \tau} \frac{\tau_j|\langle n_j, \tau_j|1\rangle|^2}{\partial_\tau \mu_{n_j}(\tau_j)(\tau' - \tau_j)}\right) + \sum_{j, \tau_j \le \tau} \frac{\tau_j|\langle n_j, \tau_j|1\rangle|^2}{\partial_\tau \mu_{n_j}(\tau_j)(\tau' - \tau_j) \pm i0^+}\right\}\right]. \tag{56}$$

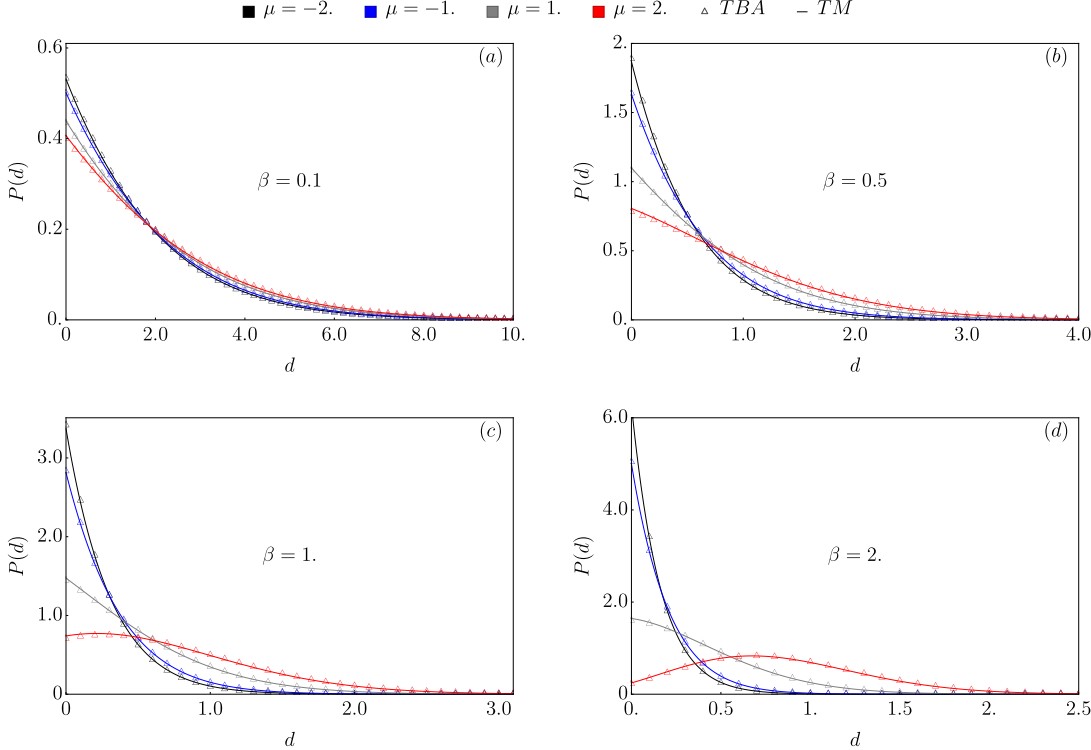

Figure 4: Comparison between probability distribution for the particle number density $P(d) \equiv \mathbb{P}(|\phi|^2 = d)$ on thermal states computed extracting the root density from TBA equations (37) and using the Transfer Matrix approach. Here $\beta$ is the inverse temperature characterising the state.

Above, we summed and substracted the singular part, letting $\tau_j$ be the zeroes of the $n_j^{\text{th}}$ eigenvalue $\mu_{n_j}(\tau_j) = 0$. The first term is regular and real, the second term is easily integrated leading to (we focus on the contribution of a single pole)

$$\exp\left[-\frac{16}{\pi}\int_0^\tau \mathrm{d}\tau' \frac{\tau_j|\langle n_j, \tau_j|1\rangle|^2}{\partial_\tau \mu_{n_j}(\tau_j)(\tau' - \tau_j) \pm i0^+}\right] =$$

$$\left|\frac{\tau - \tau_j}{\tau_j}\right|^{-\frac{16}{\pi}\frac{\tau_j|\langle n_j,\tau_j|1\rangle|^2}{\partial_\tau\mu_{n_j}(\tau_j)}} e^{\pm\frac{16}{\pi}\frac{\tau_j|\langle n_j,\tau_j|1\rangle|^2}{|\partial_\tau\mu_{n_j}(\tau_j)|}i\pi\theta(\tau-\tau_j)}. \quad (57)$$

The above expression is not explicitly real and has a non-trivial dependence on the sign $\pm$. Both issues are resolved if it holds true

$$-\frac{16}{\pi}\frac{\tau_j|\langle n_j, \tau_j|1\rangle|^2}{\partial_\tau\mu_{n_j}(\tau_j)} = 1 \quad (58)$$

for any $\tau_j$, then

$$\exp\left[-\frac{16}{\pi}\int_0^\tau \mathrm{d}\tau' \frac{\tau_j|\langle n_j, \tau_j|1\rangle|^2}{\partial_\tau\mu_{n_j}(\tau_j)(\tau'-\tau_j)\pm i0^+}\right] = \frac{\tau_j - \tau}{\tau_j}, \quad (59)$$

which is real and analytic and does not depend on the sign of the regularization. The identity Eq. (58) can be analytically proven, as we show in Appendix C. Employing Eq. (58), we are

led to a well-defined expression for the $F$ function

$$F(-\tau^2) = \exp\left[-\int_0^\tau d\tau'\left(\frac{16}{\pi}\sum_n \frac{\tau'|\langle n,\tau'|1\rangle|^2}{\mu_n(\tau')} + \sum_{j,\tau_j\leq\tau} \frac{\partial_\tau\mu_{n_j}(\tau_j)}{\partial_\tau\mu_{n_j}(\tau_j)(\tau'-\tau_j)}\right)\right]\prod_{j,\tau_j\leq\tau}\frac{\tau_j-\tau}{\tau_j}.$$
(60)

In Fig. 4 we evaluate the exact result (48) on thermal states and compare it with the numerical data obtained with the TM approach, finding perfect agreement. However, let us stress once again that Eq. (48) is of much wider applicability and holds true on arbitrary GGEs, hence can be used in out-of-equilibrium protocols as well, as we will do in Sec 5.

**The semiclassical limit in the lab.** Along this paper, we resided to the small parameter $\hbar$ as a convenient way to take the semiclassical limit. While this is probably the most natural approach, on the other hand it is not of immediate experimental applicability: however, as we have already commented at the beginning of this section, the same limit can be analogously obtained in a high temperature and weak coupling limit with $\hbar$ kept constant, where the quantum temperature and the interaction are sent to zero while keeping their ratio fixed. More specifically, let us consider the Lieb-Liniger Hamiltonian Eq. (2) with the following parametrization ($g = \kappa\hbar^4$ in Eq. (2))

$$\hat{H} = \int dx \left[\frac{\hbar^2}{2m}\partial_x\psi^\dagger\partial_x\psi + g\,\psi^\dagger\psi^\dagger\psi\psi\right],$$
(61)

and let us consider a thermal state $e^{-\beta_q(\hat{H}-\mu\hat{N})}$, then the classical thermal state Eq. (24) is achieved in the $\beta_q, g \to 0$ limit, provided we keep the ratio fixed in such a way that the classical temperature $\beta$ is fixed $\beta = \beta_q/((2m\hbar^{-2})g)$. In this case, the quantum expectation values go to the classical ones through the dictionary $\psi \to \phi/\sqrt{2m\hbar^{-2}g}$. The semiclassical limit of thermal states and the corrections induced by quantum fluctuations are well controlled (see eg. Ref [109]) and out-of-equilibrium setups in the weakly interacting and high density regime are expected to be well described by the classical physics as well.

## 4 The classical integrability of the NLS: the inverse scattering method

In this section, following Refs. [38, 39, 49], we provide a brief summary of the classical integrability of the NLS equation. The main aim is to put in evidence the connection with the semiclassical limit of the quantum problem and to find an efficient procedure to extract the classical root density from the initial state. For certain initial conditions with zero energy density, see e.g. Refs [110, 111], the inverse scattering problem can be completely solved and the whole time evolution predicted. On the other hand, as we extensively commented, our main focus is on initial conditions with a well-defined thermodynamic limit, i.e. with a finite energy density, and in the determination of the final steady state.

**Zero curvature condition.** The remarkable observation which led to the development of the inverse scattering trasform is that the evolution equation can be seen as a compatibility condition for a linear scattering problem depending parametrically on the spectral parameter $\lambda$. More specifically, for the NLS model, one introduces the following system

$$\partial_x F(t,x) = U_\lambda(t,x)F(t,x),$$
(62a)

$$\partial_t F(t,x) = V_\lambda(t,x)F(t,x),$$
(62b)

where $U_\lambda$, $V_\lambda$ depend on the spectral parameter $\lambda$ and on the field $\phi(t,x) = u(t,x) + \iota v(t,x)$. Expressed in terms of standard Pauli matrices, they read (we omit the explicit space-time dependence to simplify the notation)

$$U_\lambda = -\frac{\iota \lambda \sigma_z}{2} + \frac{1}{\sqrt{2}}(\sigma_x u + \sigma_y v), \tag{63a}$$

$$V_\lambda = -\frac{\iota \sigma_z (\lambda^2 + |\phi|^2)}{2} - \frac{1}{\sqrt{2}}\left[\sigma_x(\lambda u + \partial_x v) + \sigma_y(\lambda v - u_x)\right]. \tag{63b}$$

The two-component field $F(t,x)$ can be expressed in terms of the eigenstates of $\sigma_z$ as

$$F(t,x) = f_1 |\!\uparrow\rangle + f_2 |\!\downarrow\rangle \ . \tag{64}$$

Notice that the system of the two differential equations (62a) and (62b) are compatible only if it holds the so called zero curvature condition

$$\partial_t U_\lambda - \partial_x V_\lambda + [U_\lambda, V_\lambda] = 0 \ . \tag{65}$$

This equation has a clear geometrical interpretation in terms of gauge fields. Indeed, let's define the covariant derivative as

$$D_0 = \partial_0 - A_0 \,, \qquad D_1 = \partial_1 - A_1 \,, \tag{66}$$

where $\partial_0 = \partial_t$, $\partial_1 = \partial_x$, while $A_0 = V$ and $A_1 = U$ are the components of a gauge potential. Hence, Eq. (65) is equivalent to

$$[D_0, D_1] = 0 \ . \tag{67}$$

Consider the differential form $A = A_\mu dx^\mu = V_\lambda dt + U_\lambda dx$ and a closed path $\Gamma$ in space-time. In light of Eq. (65), the differential form $A$ is a closed form and an application of Stokes theorem gives

$$\mathcal{P} \exp\left(\oint_\Gamma A_\mu dx^\mu\right) = \mathcal{P} \exp\left(\int_\Gamma DA\right) = \mathbb{I}, \tag{68}$$

where $\mathcal{P}$ is the path ordering operator. The above operator implements the parallel transport, which is trivial on a closed path because of the vanishing curvature.

**Monodromy matrices.** Let's now consider the closed path defined by the points $(t_1, x)$, $(t_1, x)$, $(t_2, x)$, $(t_2, x)$ and $(t_1, x)$ (see Fig. 5) and define the propagators (monodromy matrices),

$$T_\lambda(t; x, y) = \mathcal{P} \exp\left(\int_x^y U_\lambda(t, x') dx'\right), \quad S_\lambda(t_1, t_2; x) = \mathcal{P} \exp\left(\int_{t_1}^{t_2} V_\lambda(t', x) dt'\right). \tag{69}$$

Using (68) and (69) we can write,

$$S_\lambda(t_2, t_1; x) T_\lambda(t_2; y, x) S_\lambda(t_1, t_2; y) T_\lambda(t_1; x, y) = \mathbb{I} \ . \tag{70}$$

From the elementary properties $S_\lambda^{-1}(t_1, t_2; x) = S_\lambda(t_2, t_1; x)$ and $T_\lambda^{-1}(t; x, y) = T_\lambda(t; y, x)$ one has

$$T_\lambda(t_1; x, y;) = S_\lambda^{-1}(t_1, t_2; y) \, T_\lambda(t_2; x, y) S_\lambda(t_1, t_2; x). \tag{71}$$

Hence, if there exist two points in the space-time such that

$$V_\lambda(t, x) = V_\lambda(t, y) \,, \tag{72}$$

then, in view of the relation (71), the quantity

$$\mathcal{T}_\lambda(x, y) = \mathrm{Tr}[T_\lambda(t; x, y)] \tag{73}$$

is time-independent. The simplest ways to satisfy Eq. (72) are to take

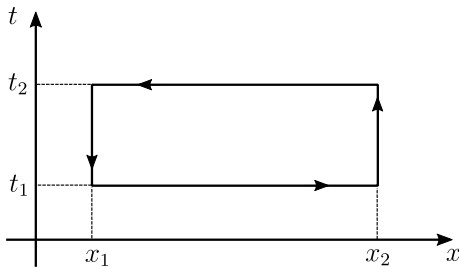

Figure 5: *Closed path relative to the definition of the monodromy matrices Eq.* (69).

- periodic boundary conditions, i.e. $\phi(t, x + L) = \phi(t, x)$;

- open boundary conditions, i.e. $\phi(t, 0) = \phi(t, L) = 0$.

Note that, for a given configuration of the field $\phi(t_0, x)$ at given time $t = t_0$, the propagator $T_\lambda(t_0; x, y)$ can be efficiently computed numerically by simply discretizing the path-ordered integral in (69) and expressing it in terms of the following matrix product

$$T_\lambda(t_0; x, y) \overset{\Delta x \to 0}{=} W_{\Delta x}(t, x) W_{\Delta x}(t, x + \Delta x) \dots W_{\Delta x}(t, y - \Delta x), \quad W_{\Delta x}(t, x) \equiv e^{\Delta x U_\lambda}. \tag{74}$$

Expanding Eq. (71) for $t_1 \to t_2$, one sees that $T_\lambda(t; x, y)$ and $M(\lambda) = V_\lambda(t, x)$ form the so-called Lax pair

$$\dot{T}_\lambda = [M, T_\lambda]. \tag{75}$$

Then for any value of the spectral parameter $\lambda$, the quantity defined in Eq. (73) can be used to generate an infinite set of conserved quantities.

**Thermodynamic limit.** As explained in Sec. 2, in the thermodynamic limit the stationary states of the Lieb-Liniger (and correspondingly of the NLS) model is completely described in terms of the root density. We already mentioned that via Eq. (12), the root density is in one-to-one correspondence with the expectation values of the conserved quantities. Since, in the semiclassical limit, the conserved quantites are all generated by Eq. (73), it is natural to expect a relation between $\mathcal{T}_\lambda(x, y)$ and the root density. In order to unveil it, we first consider the system on a finite volume of size $L$ and then take the limit $L \to \infty$ at finite energy density. We thus enforce periodic boundary conditions for $x \in [0, L]$: then Eq. (71) implies $\mathcal{T}_\lambda(0, L)$ is conserved for any $\lambda$. By performing a series expansion at large $\lambda$ [38, 48, 112], one can show indeed that

$$\mathcal{T}_\lambda(0, L) \overset{L \to \infty}{=} 2 e^{L \pi \rho(\lambda)} \cos(L \Omega(\lambda) - L \lambda / 2), \tag{76}$$

where $\Omega(\lambda)$ is the generating function of all local conserved densities via

$$\Omega(\lambda) = \sum_{n=0}^{\infty} \frac{I_n}{\lambda^n} \quad , \quad I_n \equiv \int d\lambda \, \rho(\lambda) \lambda^n. \tag{77}$$

Now, since $\sigma_x U_\lambda \sigma_x = U_\lambda^*$, the diagonal entries of $\mathcal{T}_\lambda(0, L)$ are complex conjugate. Combining this fact with Eq. (76), we deduce that the top-left component of the propagator has the form[1]

$$[\mathcal{T}_\lambda(0, L)]_{1,1} \sim e^{L(\pi \rho(\lambda) + \iota \Omega(\lambda) - \iota \lambda / 2)}. \tag{78}$$

---

[1]This corresponds to the propagator of NLS problem defined on the infinite line and vanishing at infinity [38]. Up to boundary terms, which are irrelevant in the thermodynamic limit we are considering, this can always be obtained embedding a periodic configuration defined for $x \in [0, L]$ inside the whole real line and setting it to zero outside.

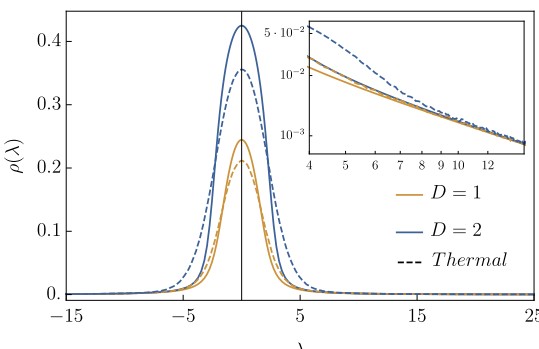

Figure 6: Root density generated in two different quenches (continue lines) compared with the thermal state it would be reached in the absence of integrability (dashed lines). More specifically, the system is prepared in a non-interacting thermal state with Hamiltonian $H_{t<0} = \int dx \, |\partial_x \phi|^2$ and inverse temperature $\beta = 1$ and densities chosen to be in the strong-interacting regime $D = \langle |\phi|^2 \rangle$ equal to 1 and 2 respectively. The parameters of the putative thermal states shall be chosen to match the initial value of energy and density. Note that because of the UV divergence of the energy, the final value of $\beta = 1$ remains unchanged, while $\mu$ is adjusted to match the initial densities. Inset: tails of the roots in logarithmic scale, which besides of collapsing on the same curve they also agree with the initial thermal distribution, as we explain in the main text.

**Post-quench root density.** In the quench protocol we have considered in this paper (see next Section for more details), the system is initially prepared in a thermal state and we are interested in the post-quench root density which provides its GGE representation. This can be achieved

1. drawing a field configuration $\phi(x)$ from the Boltzmann measure given in Eq. (24);

2. computing the matrix $\mathcal{T}_\lambda(0, L)$ via eq. (74);

3. using Eq. (76) to extract the root density as

$$\rho(\lambda) = \lim_{L \to \infty} \frac{2 \log |[\mathcal{T}_\lambda(0, L)]_{1,1}|}{\pi L} \ . \tag{79}$$

Note that the root density is self-averaging and therefore for sufficiently large $L$, a single configuration would be sufficient. In practice, for the numerical implementation, we have averaged Eq. (79) over many field configurations so that finite size fluctuations were suppressed (see below).

## 5 Benchmark of the analytic results: time evolution vs GGE predictions

In this section we compare the analytical findings of the previous section with ab-initio numerical simulations of the NLS (details in App. D). In the quantum quench protocol, the initial state is often chosen in the form of a pure state in order to enhance its quantum properties, but obviously density matrixes (such as thermal states or GGEs) are equally physically-motivated states. In the perspective of using the classical theory as an approximation of the quantum

model, we have initialized the system directly in a Gibbs ensemble of the NLS and we have let the system evolve with a different interaction for $t > 0$. In order for the GGE prediction to emerge, an averaging procedure is required: within the quantum context, the expectation value on the initial state accounts for both quantum and statistical fluctuations, while in the classical case the first are obviously absent. Hence, we average with respect to the initial random configurations. More specifically, the microscopic protocol consists in three steps:

- Sample the initial field configuration from the desired thermal ensemble.

- Evolve each field configuration with the deterministic NLS equation, measuring the observables along the evolution.

- Repeat the above steps for several initial field configurations and average over them.

As stated before, in the thermodynamic limit, the spatial average of a single field configuration would be sufficient. In practice, as we work at finite volume, we average over the initial ensemble to further suppress the fluctuations. The numerical computation of the root density on the initial state, which is then fed to the expressions of Section 3 to access the steady-state's observables, is performed in similar steps. For any field configuration dragged from the initial ensemble, we compute the associated transfer matrix through a brute-force computation of Eq. (74), then the root density is extracted by means of Eq. (79) computed at finite volume: in order to extract the thermodynamic limit, we use the self-averaging property of the system and average the root density over the initial configurations (see App.D for further details). The averaged root density is then used in computing the moments of the density and the whole FCS. In the following, we start from non-interacting thermal states based on $H_{t<0} = \int dx \, |\partial_x \phi|^2$ and then evolve with Eq. (4). Thermal states in free systems (and, more generally, GGEs) are easily generated in the Fourier space, where the modes are independently gaussianly distributed (see App. D). The sampling of an interacting thermal states can be achieved with a Metropolis-Hasting algorithm [113], similar to what it has been done in Ref. [81]: however, we do not expect any qualitatively-different behavior, hence we consider the non-interacting case for the sake of simplicity.

In Fig. (6) we compare the root densities obtained through different quenches. More specifically, the system is initialized in a non-interacting thermal state and then let evolve in the presence of interactions. We point out that at large rapidities the root density is expected to coincide both with the putative interacting thermal state, as well as the non interacting thermal state. This can be forecast in view of the fact that large rapidities are weakly interacting, hence they are not affected by changes of the interaction.

In Fig. 7 we show the time evolution of the density's moments for different choices of the initial state, finding perfect agreement with relaxation to the forecast GGEs. Instead, in Fig. 8 we focus on the time-evolution of the full counting statistics: the time-evolved distribution clearly approaches the GGE prediction, confirming once again the validity of our method. The non-monotonous behavior of the FCS unveils the highly interacting nature of the system. Indeed, in the absence of the interactions and thus on a gaussian steady state, the FCS is straightforwardly shown to be a simple exponential $P(d) = D^{-1}e^{-d/D}$ (with $D$ the average density): in Fig. 8 at time $t = 0$ the system is initialized in a gaussian ensemble and the curves are perfect exponentials. Then, at later times the effect of interactions becomes more and more important, shifting the peak of the distribution from $d = 0$ to a finite value. At equilibrium and for finite intervals, this features have been extensively studied, for example, in Ref. [75].

This classical out-of-equilibrium protocol can be viewed as an approximation of a quench in the quantum Lieb-Liniger, from zero to a small value of the interaction and starting from a high temperature thermal state. Quenches starting from weakly interacting initial thermals states, which are approximated by interacting classical thermal ensembles, can be considered as well and are a simple application of our general method.

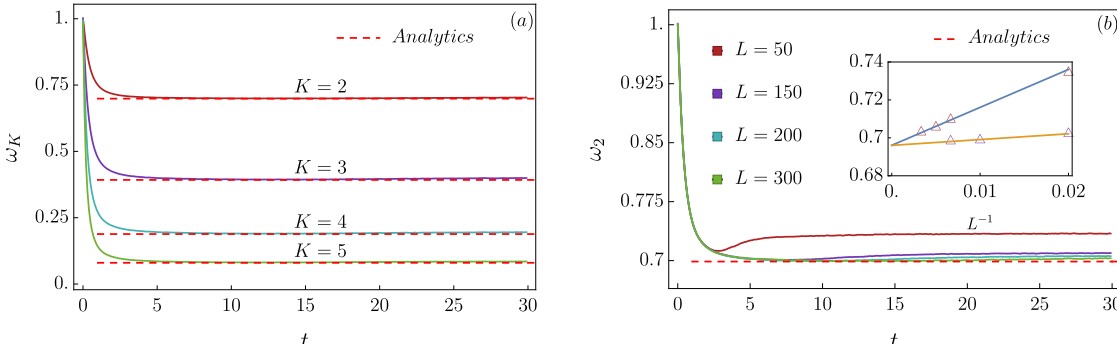

Figure 7: (a) Comparison between time evolution of rescaled density moments $\omega_K = \langle|\phi|^{2K}\rangle/[(\langle|\phi|^2\rangle)^K K!]$ computed from the direct simulation of the NLS and steady state values predicted from Eq. (39) (red dashed lines). The initial state in chosen as in Fig.6 and we focus on the $D = 2$ case. Since the initial state is gaussian, at time $t = 0$ we have $\omega_K = 1$, however as time passes the strongly interacting nature of the system is unveiled and the values of $\omega_K$ depart from unity, relaxing to the predicted asymptotic value with excellent accuracy. In our microscopic simulation we experienced important finite-size effects and the data shown in the plot are extrapolated to the thermodynamic limit. (b) We provide a brief analysis of the finite-size effects focusing on the time evolution of the first non trivial observable $\omega_2$. Inset: extrapolation to the thermodynamic limit of the plateau extracted from the microscopic evolution (blue line) and of the plateau obtained from the root density (yellow line). In both cases, we observe a clear $\propto L^{-1}$ approach to the asymptotic values, which are in perfect agreement.

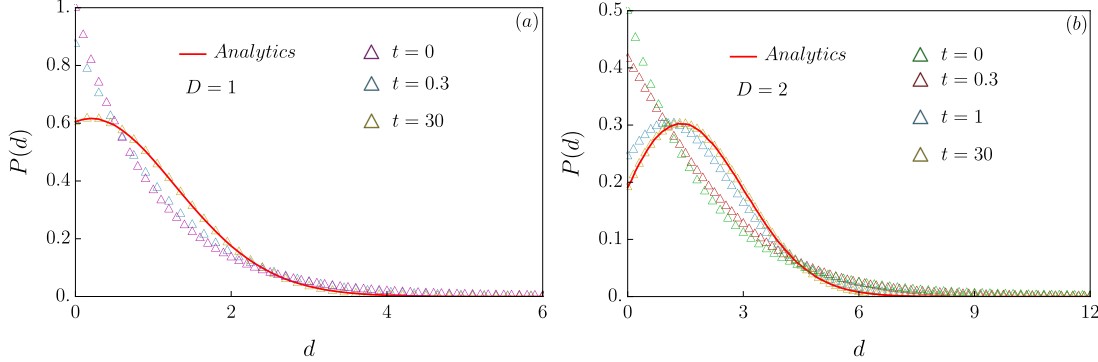

Figure 8: For the same protocol described in Fig. 7, we provide the time evolution of the FCS of the density operator. Its profile is numerically computed at different time steps and at late time clearly collapses on the analytical prediction Eq. (48), where we employed the GGE root density numerically extracted from the initial conditions.

## 6 Conclusions

In this paper we thoroughly analyzed quench protocols in the classical Non-Linear Schröedinger equation, which can be viewed as the semiclassical limit of the interacting Bose gas. Our interest was twofold: on the one hand, we aimed to study the classical model taking advantage of existing results on the quantum model, projected in the classical world through a proper

semiclassical limit. To this end, we determined new exact expressions for the moments of the local density in the form of recursive integral equations, which are valid for arbitrary GGEs. These expressions are the classical analogue of the study of Ref. [46, 47]. Moreover, viewing the classical NLS as the non relativistic limit of the classical sinh-Gordon model, we provided the exact probability distribution of the local density: this new result has no quantum analogue so far.

On the other hand, we were interested in investigating the quantum model within the semiclassical regime: a key-point of addressing a quench protocol is being able to determine the emergent GGE from the initial conditions and, within the quantum case, there are no analytical or numerical methods to answer this question (with a few remarkable exceptions [70, 72]). We completely solve this problem in the classical regime, providing an efficient numerical procedure to determine the GGE from the initial state.

Several interesting questions naturally emerge from our investigation. First of all, even though we mainly focused on the NLS, the same approach can be applied to study other classical integrable systems. In particular, it would be interesting to address systems whose TBA possess more than a single species of excitation (string) and lattice systems, such as the Toda lattice [38]. It would be interesting to gain further insight in the semiclassical limit of other relevant quantum models, such as the XXZ spin chain and its classical counterpart, the Landau-Lifshitz model [114].

Hybrid semiclassial-quantum methods could be imagined as well, with the semiclassical computation of the initial root density, which can then be used as an initial condition for quantum methods, for example the generalised hydrodynamics [83, 84]. This could be implemented, for instance, to study the the famous Newton-Cradle experiment [9]: the state after the initial Bragg pulse still lacks a faithful description [115].

Lastly, it would be very intriguing to improve the semiclassical approximation of the quantum model: the classical system can be regarded as the zeroth order in the $\hbar$ expansion of the quantum one. Is it possible to include further orders in $\hbar$, providing an expansion of the quantum model building on the classical theory? Can we include these quantum corrections in our method to determine the root density? We leave these exciting questions open for future investigations.

# Acknowledgements

G.D.V.D.V. is indebted with Sergio Caracciolo and Benjamin Doyon for their support during this research. We are thankful to O. Gamayun and Y. Miao for useful discussions on the classical integrability of the NLS equation.

**Funding information**   A.B. acknowledges the support from the European Research Council under ERC Advanced Grant No. 743032 DYNAMINT.

# A   The transfer matrix method

The transfer matrix method takes advantage of the classical/quantum correspondence of the partition functions to map the 1d classical problem in a 0d quantum one, reducing the problem of computing classical thermal averages of local observables to find the ground state of a two-dimensional Hamiltonian [48, 116–119]. For the sake of clarity, we consider directly the momenta of the density, but the same procedure can be repeated for any observable. The

classical expectation value is given by

$$\langle|\phi(0)|^{2n}\rangle = \frac{1}{\mathcal{Z}} \int \mathcal{D}\phi \, |\phi(0)|^{2n} e^{-\beta \int_0^L dx \, |\partial_x \phi|^2 + |\phi|^4 - \mu |\phi|^2} . \tag{80}$$

Lately, we will consider the limit $L \to \infty$. We rewrite the above as a quantum expectation value introducing a two-dimensional quantum-mechanical Hilbert space $\mathcal{H}_{TM}$ spanned by wavefunctions having support on $(u, v) \in \mathbb{R}^2$, together with the quantum Hamiltonian $H_{TM}$

$$H_{TM} = -\frac{1}{4}\beta^{-1}(\partial_u^2 + \partial_v^2) + \beta(u^2 + v^2)^2 - \beta\mu(u^2 + v^2). \tag{81}$$

Then it is a straightforward excercise to establish the correspondence between the classical expectation values and those in the ancillary quantum problem

$$\langle|\phi(0)|^{2n}\rangle = \frac{\mathrm{Tr}_{\mathcal{H}_{TM}}\left[|u+iv|^{2n}e^{-LH_{TM}}\right]}{\mathrm{Tr}_{\mathcal{H}_{TM}}\left[e^{-LH_{TM}}\right]} = \frac{\sum_j \langle j||u+iv|^{2n}|j\rangle e^{-LE_j}}{\sum_j e^{-LE_j}}, \tag{82}$$

where $|j\rangle$ are the eigenvectors of the quantum Hamiltonian $H_{TM}|j\rangle = E_j|j\rangle$. Above, the trace appears because of the spatial periodicity of the classical model: the length $L$ plays the role of an effective inverse temperature for the auxiliary quantum problem. If we are now interested in the thermodynamic limit we let $L \to \infty$, hence the quantum trace is projected on the ground state. Thus, we finally have

$$\langle|\phi(0)|^{2n}\rangle_{TDL} = \int du dv |\Psi_G(u,v)|^2 |u+iv|^{2n}, \tag{83}$$

where $\Psi_G$ is the ground state wavefunction of the Hamiltonian $H_{TM}$. Rather than approaching directly the two dimensional problem, it is convenient to use the rotation symmetry in terms of polar coordinates $u = r\cos\theta$ and $v = r\sin\theta$

$$H_{TM} = -\frac{1}{4r^2}\partial_\theta^2 - \frac{1}{4\beta r}\partial_r(r\partial_r) + \beta r^4 - \beta\mu r^2. \tag{84}$$

The ground-state wavefunction is symmetric under rotations, hence it is independent on $\theta$ and as such it can be found (numerically) solving the problem in the single-variable $r$.

## B  Derivation of Eq. (43)

In this appendix we derive Eq. (43), which is the first building block to access the FCS. The starting point is the classical ShG model: in contrast with the quantum Hamiltonian (1), we remove the $\hbar$ coupling and set the mass scale $m_0 = 1/2$. Therefore, we consider the classical Hamiltonian

$$H_{\mathrm{ShG}} = \int dx \left[\frac{1}{2c^2}\partial_t\Phi^2 + \frac{1}{2}(\partial_x\Phi)^2 + \frac{c^2}{4g^2}(\cosh(g\Phi) - 1)\right], \tag{85}$$

where $\Pi$ and $\Phi$ are classical conjugated fields $\{\Phi(x), \Pi(y)\} = \delta(x-y)$. Within the quantum case, the non-relativistic limit has been thoroughly examined to study the LL model starting from known results in the ShG field theory [41, 42, 120]. Indeed, the result for the one-point functions of Ref. [46, 47] has been obtained taking the NR limit of a result of Negro and Smirnov [121, 122] (see also Ref. [123]) in the quantum ShG. Here, we follow a similar procedure, albeit in the classical case.

The classical analogue of the Negro-Smirnov formula has been derived in Ref. [81] and it consists in a recursive set of integral equations for the expectation values of certain operators in the ShG model, the formula being valid for arbitrary GGEs

$$\frac{\langle e^{(k+1)g\Phi}\rangle}{\langle e^{kg\Phi}\rangle} = 1 + (2k+1)\frac{cg^2}{4\pi}\int d\theta \, e^\theta \vartheta_{\text{ShG}}(\theta)p_k(\theta). \tag{86}$$

Above, $\vartheta_{\text{ShG}}(\theta)$ is the filling fraction in the classical ShG theory and $p_k(\theta)$ satisfies

$$p_k(\theta) = e^{-\theta} + \frac{cg^2}{4}\fint \frac{d\theta'}{2\pi}\frac{1}{\sinh(\theta-\theta')}(2k-\partial_{\theta'})[\vartheta_{\text{ShG}}(\theta')p_k(\theta')]. \tag{87}$$

Presenting a detailed analysis of the NR limit goes beyond the aim of this work, which is mostly focused on the semiclassical case: we leave to the original references [41, 42] the details (even though they concern the limit of the quantum model, the analysis in the classical case is exactly the same) and quote the key relations. The NR limit is attained sending $c \to \infty$, while instead $g \to 0$. In particular, the NLS model (3) is attained if one sets

$$g^2 = 16/c^2. \tag{88}$$

At the level of TBA, one connects the relativistic rapidity $\theta$ and the Galilean one $\lambda$ looking at the momentum eigenvalue $\lambda = \frac{c}{2}\sinh\theta$: imposing $\lambda$ to be finite in the $c \to \infty$ limit, one must consider small $\theta$−rapidities, therefore at first order one simply has $\theta \simeq 2\lambda c^{-1}$. Finally, the filling in the ShG simply goes to the NLS one, provided we correctly link the rapidities

$$\vartheta(\lambda) \simeq \vartheta_{\text{ShG}}(2\lambda c^{-1}). \tag{89}$$

The last ingredient we need is the mapping at the level of observables, which is

$$\lim_{c\to\infty}\langle\Phi^{2n+1}\rangle = 0 \qquad \lim_{c\to\infty}\langle\Phi^{2n}\rangle = \binom{2n}{n}\langle|\phi|^{2n}\rangle. \tag{90}$$

We can now proceed and take the NR limit of Eq. (86), from which Eq. (43) will follow. Rather than naively take $c \to \infty$ (which would lead to a trivial result), we first rescale $q = kc^{-1}$ and take the NR limit keeping $q$ fixed. Looking at the l.h.s. of Eq. (86) (and rescaling the coupling $g$ as per Eq. (88)), one gets

$$\frac{\langle e^{(q+c^{-1})4\Phi}\rangle}{\langle e^{q4\Phi}\rangle} \simeq 1 + \frac{1}{c}\partial_q\log\Big[\lim_{c\to\infty}\langle e^{q4\Phi}\rangle\Big] + \mathcal{O}(c^{-2}). \tag{91}$$

We retain up to first order in $c$: the $\mathcal{O}(c^0)$ term in the above exactly cancels the first term in the r.h.s. of Eq. (86) and one should compare the next order. By means of a direct power expansion and using Eq. (90), one readily gets

$$\lim_{c\to\infty}\langle e^{q4\Phi}\rangle = \sum_n q^{2n}\frac{(16)^n}{(n!)^2}\langle|\phi|^{2n}\rangle, \tag{92}$$

which is the l.h.s. of Eq. (43). The r.h.s. is readily found taking the NR limit of Eq. (86) (and Eq. (87): while doing so one defines $W_q(\lambda) = \lim_{c\to\infty}(p_{cq}(2\lambda c^{-1}))$, then integrating in $q$ and inverting the logarithm appearing in Eq. (91). Eq. (43) then readily follows.

## C  Proof of Eq. (58)

In this appendix, we show that Eq. (58) is verified. First of all, by Hellman-Feynman theorem, we observe that

$$\partial_\tau \mu_{n_j}(\tau) = \langle n, \tau | \partial_\tau \Omega_\tau | n, \tau \rangle = \frac{-4\imath}{\pi} \oint d\lambda d\lambda' \frac{\varphi^*_{n,\tau}(\lambda)\varphi_{n,\tau}(\lambda')}{\lambda - \lambda'}, \tag{93}$$

where we set $\varphi_{n,\tau}(\lambda) = \langle \lambda | n, \tau \rangle$. Now, since $|n, \tau\rangle$ is an eigenvector of $\Omega_\tau$, we have the equation

$$\vartheta(\lambda)^{-1}\varphi_{n,\tau}(\lambda) - \oint \frac{d\lambda'}{\pi}\left[\frac{4\imath\tau\varphi_{n,\tau}(\lambda') - \varphi'_{n,\tau}(\lambda')}{\lambda - \lambda'}\right] = \mu_{n,\tau}\varphi_{n,\tau}(\lambda). \tag{94}$$

Now, we multiply this equation times $\lambda\varphi^*_{n,\tau}(\lambda)$, take the imaginary part and integrate over $\lambda$. The first term as well as the right hand side give vanishing contribution, as $\vartheta(\lambda)$ and $\mu_{n,\tau}$ are real. We are thus left with

$$\Im \int d\lambda \, \lambda\varphi^*_{n,\tau}(\lambda) \oint \frac{d\lambda'}{\pi}\left[\frac{4\imath\tau\varphi_{n,\tau}(\lambda')}{\lambda - \lambda'}\right] = \Im \int d\lambda \, \lambda\varphi^*_{n,\tau}(\lambda) \oint \frac{d\lambda'}{\pi}\left[\frac{\varphi'_{n,\tau}(\lambda')}{\lambda - \lambda'}\right]. \tag{95}$$

From the left hand side, we get

$$\int d\lambda \, d\lambda'\left[\frac{4\tau\lambda(\varphi^*_{n,\tau}(\lambda)\varphi_{n,\tau}(\lambda') + \varphi^*_{n,\tau}(\lambda')\varphi_{n,\tau}(\lambda))}{\lambda - \lambda'}\right] = 4\tau\left|\int d\lambda\varphi_{n,\tau}(\lambda)\right|^2 = 4\tau|\langle 1|n, \tau\rangle|^2, \tag{96}$$

where in the second equality we exchanged the variables $\lambda \leftrightarrow \lambda'$. Similarly, from the right-hand side, integrating by part twice we find

$$\Im \int d\lambda \, d\lambda'\left[\frac{\lambda\varphi^*_{n,\tau}(\lambda)\varphi'_{n,\tau}(\lambda')}{\lambda - \lambda'}\right] = -\Im \int d\lambda \, d\lambda'\left[\frac{\partial_\lambda(\lambda\varphi^*_{n,\tau}(\lambda))\varphi_{n,\tau}(\lambda')}{\lambda - \lambda'}\right] =$$
$$= \int d\lambda \, d\lambda'\left[\frac{\imath\varphi^*_{n,\tau}(\lambda)\varphi_{n,\tau}(\lambda')}{\lambda - \lambda'}\right] - \Im \int d\lambda \, d\lambda'\left[\frac{\lambda\varphi'^*_{n,\tau}(\lambda)\varphi_{n,\tau}(\lambda')}{\lambda - \lambda'}\right]. \tag{97}$$

The last term can be easily recast to be the same as the left-hand side up to a minus sign. We thus obtain

$$4\tau|\langle 1|n, \tau\rangle|^2 = \int d\lambda \, d\lambda'\left[\frac{\imath\varphi^*_{n,\tau}(\lambda)\varphi_{n,\tau}(\lambda')}{\lambda - \lambda'}\right] = -\frac{\pi}{4}\partial_\tau\mu_{n_j}(\tau), \tag{98}$$

where in the last equality, we used (93). This last equation is clearly equivalent to (58).

## D  Numerical Methods

Here we explain in detail the numerical procedure used for the ab-initio simulations of the classical model. The dynamics of the system is fully deterministic and ruled by the PDE (4). Initial field configurations are randomly generated according to a given probability distribution, which is chosen stationary with respect to the non-interacting dynamics. In other words, we sample the initial field configurations from non-interacting thermal states and, more generally, GGEs. The root density can be easily numerically extracted from the initial state in view of Eqs. (74) and (79). We now discuss in details each step of the algorithm.

## D.1 Solution of the equation of motion

We now consider the problem of solving the equation of motion starting from a given initial field configuration

$$\begin{cases} i\partial_t \phi(t,x) = -\partial_x^2 \phi(t,x) + 2|\phi(t,x)|^2 \phi(t,x) \\ \phi(0,x) = \phi_0(x) \end{cases}. \tag{99}$$

We approach the problem through a Hamiltonian-splitting procedure [124], which is stable and exactly conserves the number of particles. We split the Hamiltonian $H$ in three parts

$$H = H_1 + H_2 + H_3, \tag{100}$$

with

$$H_1 = H_3 = |\phi|^2 \phi, \quad H_2 = -\partial_x^2. \tag{101}$$

The field is discretized as $\phi_{j,i} \equiv \phi(j\Delta t, i\Delta x)$ with $i = 0, \ldots, N-1$ and $j = 0, \ldots, M-1$. Then, each step in the time evolution is performed through three different phases

$$\phi_{j+1/3,i} = \sum_{i'} \left[ e^{-iH_1\Delta t} \right]_{i,i'} \phi_{j,i'}, \tag{102}$$

$$\phi_{j+2/3,i} = \sum_{i'} \left[ e^{-iH_2\Delta t} \right]_{i,i'} \phi_{j+1/3,i'}, \tag{103}$$

$$\phi_{j+1,i} = \sum_{i'} \left[ e^{-iH_3\Delta t} \right]_{i,i'} \phi_{j+2/3,i'}. \tag{104}$$

The first and third steps are easily performed in the real space, while the one in the middle is performed in the Fourier space, where $H_2$ acts diagonally.

$$\tilde{\phi}_{j+2/3,k} = e^{-i\Delta t E_k} \tilde{\phi}_{j+1/3,k}, \quad E_k = \frac{2}{\Delta x^2} \left( 1 - \cos\left( \frac{2\pi}{N} k \right) \right). \tag{105}$$

In the limit $N \to +\infty$, $\Delta x \to 0$ with $L \equiv N\Delta x$ fixed, defining $\lambda \equiv \frac{2\pi}{L} k$, we have $E_k \to E(\lambda) = \lambda^2$ at leading order.

The energy scale, dominated by the kinetic term, sets the allowed values of $\Delta t$, which must always fulfill

$$\Delta t \max_k E_k \ll 1 \qquad \Rightarrow \qquad 4\Delta t \Delta x^{-2} \ll 1. \tag{106}$$

In our simulations, we always used $(\Delta x)^{-2}\Delta t \simeq 10^{-3}$. In order to extract the continuum limit, we extrapolate the finite-size data according with a simple Taylor-expansion ansatz in the length of the system

$$f(t, 1/L) = f(t,0) + \frac{1}{L} f'(t,0) + \frac{1}{2L^2} f''(t,0) + \ldots \tag{107}$$

## D.2 Generation of initial states and averages

As explained in the main text, for the sake of simplicity we focus on quenches from the free theory to the interacting one. Free thermal states (and, more in general, GGEs) are simply formulated in the Fourier space, where the modes are independent and gaussianly distributed with zero mean. Hence, field configurations are easily generated in the Fourier space and then converted back in the real one. On a thermal state with inverse temperature $\beta$ and chemical potential $\mu$, the modes are distributed according with the Raylegh-Jeans distribution

$$\langle \tilde{\phi}_k^\dagger \tilde{\phi}_q \rangle = \frac{\delta_{k,q}}{\beta \Delta x} \frac{1}{E_k - \mu}. \tag{108}$$

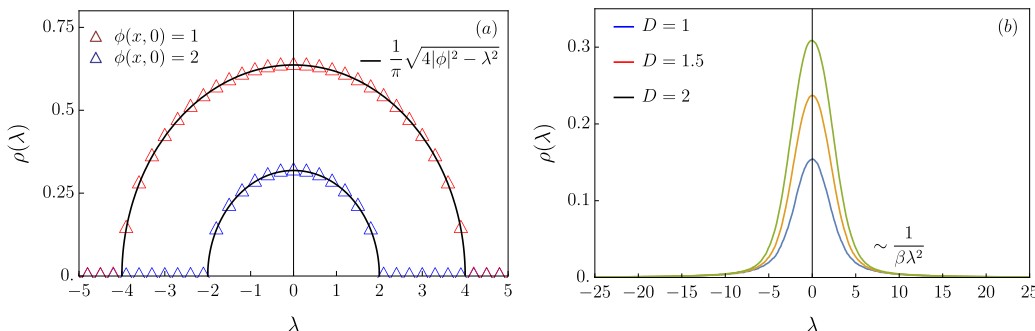

Figure 9: (a) Classical root densities for homogeneous initial field configurations. Continuous black lines are theoretical predictions Eq. (113) while the triangles are numerical samples. (b) Root densities for different values of the density of particles $D = |\phi(x)|^2$ computed with the procedure described in this Appendix and averaged over $10^4$ realizations. The field configurations are sampled from a free thermal ensemble with Hamiltonian $\int_0^L dx(|\partial_x \phi|^2 - \mu|\phi|^2)$ and inverse temperature $\beta = 0.4$. The value of the chemical potential $\mu$ is adjusted in order to obtain the desired value of the density. For large values of the rapidity, the effect of the interaction is negligible and the root density decays as $\rho(\lambda) \sim 1/(\beta\lambda^2)$.

The density of particles $D$ fixes the chemical potential according to the relation (valid in the continuum limit)

$$\mu = -\frac{1}{4\beta^2 D^2} \,. \tag{109}$$

Given an initial configuration $\mathcal{C}$ the value of a generic observable $O$ depends on it. The ensemble average is estimated as

$$\langle O \rangle = \frac{1}{N_\mathcal{C}} \sum_\mathcal{C} O[\mathcal{C}] \tag{110}$$

and analogously for the standard deviation.

### D.3 Extraction of the root density

The key identities for the extraction of the root density are Eqs. (74)-(79), which explicitly read

$$W_{\Delta x}(t,x) = \exp(\Delta x U_\lambda) = \begin{pmatrix} \cosh(\tau\Delta x) - \frac{i\lambda \sinh(\tau\Delta x)}{2\tau} & \frac{\phi^\dagger \sinh(\tau\Delta x)}{\tau} \\ \frac{\phi \sinh(\tau\Delta x)}{\tau} & \cosh(\tau\Delta x) + \frac{i\lambda \sinh(\tau\Delta x)}{2\tau} \end{pmatrix}, \tag{111}$$

where $\tau = \frac{1}{2}\sqrt{|\phi|^2 - \lambda^2}$. For a system of length $L$, the matrix $T_\lambda$ is approximated as,

$$T_\lambda \approx \prod_{i=0}^{N-1} W_{\Delta x}(t, i\Delta x), \quad N\Delta x \equiv L, \quad N \gg 1 \tag{112}$$

and the root density is easily extracted. The root density is then used as an input for the exact formulae for the one-point functions Eq. (41) and the FCS Eq. (50). A case amenable of a fully analytical check is when the initial field configuration is homogeneous in space. In this situation the matrix $T_\lambda$ does not depend on $x$ and a straightforward calculation shows that,

$$\rho(\lambda) = \frac{1}{2\pi}\sqrt{4|\phi|^2 - \lambda^2}, \tag{113}$$

which is a semicircle distribution, as we show in Fig. 9 (a). In Fig. 9 (b) we consider the numerically-extracted root density choosing as initial states free thermal states of different temperatures. We notice that this simple result is in agreement with the analytical solution of the quantum quench from the non-interacting ground state. Indeed, the root density computed in Ref. [70] collapses to the semi-circle low in the limit of small interaction, i.e. where the classical model emerges.

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
