# Peer review of "Exact out-of-equilibrium steady states in the semiclassical limit of the interacting Bose gas"

_SciPost Physics, doi:SciPost Phys. 9, 002 (2020)_

## Round 2 · Referee Report · Anonymous (Referee 1) · 2020-4-10

Strengths

1-timely
2-clear
3-relevant

Weaknesses

1- The link with the previous paper of two of the authors (ref[47]) is not made clearly. What is known from [47] and what is really new is not clear. 2- The link with results on the quenches in the quantum Lieb-Liniger model is not made.

Report

In this paper, the authors investigate out-of-equilibrium dynamics in the classical model described by the Non-Linear-Shrodinger Equation (NLSE). More precisely they are interested in the dynamics following a sudden change of the coupling constant, and they focuss on the properties of the system after relaxation. This model is integrable. It thus holds an infinite number of conserved quantities and the system does not a priory relax towards a state described by a Gibbs Ensemble. Instead, the system after relaxation is described, as long as local properties are concerned, by the set of all conserved quantities.
The quantum equivalent of the NLSE is the Lieb-Liniger (LL) model. Many progress have been done in the recent years concerning the investigation of relaxation in this model. The system, as long as local quantities are concerned, is described at long times by a Generalized Gibbs Ensemble that account for the infinite number of conserved quantities. The notion of GGE is however not very useful since the conserved quantities are not known. Moreover a truncation method fails to describe the final state in the case of a quench of the interactions. It is more advantageous to describe the system in terms of the root density function, which is equivalent to the GGE description. Many quantities can be computed knowing the root density function. In particular the zero-distance many body correlation functions can be computed. Computing the root density function after a quench is however very difficult and only few results exist. The NLSE describes the LL model in the limit hbar goes to zero. It is thus important to make the link between the quantum description and the classical one. In this paper, the authors use the results known for the LL model to learn new results about the NLSE. In turns this amounts to new results for the classical asymptotic limit of the LL model. For this purpose, the authors establish the classical counterpart of the root density function. Note that this task was already performed by some of the authors in [47], although in [47] they focuss on the Sinh-Gordon model. Then the authors derive the classical counterpart of the quantum calculation of the zero-distance many body correlation functions, namely the different moments of the density. They finally show how to use these results to compute the Full-Counting Statistic (FCS), namely the probability distribution of the density. They test those results in the case of thermal equilibrium states. In this case, the root density function is known and they compute, using their newly established formulas, the moments of the density and the FCS. They compare the results with that obtained by the Transfer Matrix Method, which applies for thermal equilibrium states. The agreement is excellent. They then apply the previous results to the case of a sudden quench of the interaction coupling constant. After relaxation, the system is solely described by its root density function and the difficult point is to compute the root density function after the quench. They use here results established in ref [47] making the link between the root density function and the conserved quantity obtained using the inverse scattering method, which requires only the knowledge of the initial field. This way, they compute the root density function after the quench. This enable them to compute the asymptotic values of the moments of the density and the FCS of the density. They compare those predictions to numerical results obtained by numerical integration of the NLSE. They find very good agreement. In their numerical tests, they use as initial state a thermal state of the model with vanishing interactions.

As an experimentalist, I am not qualified to judge the accuracy and the relevance of the theoretical derivations. The paper however is well written and can be understood by a large audience. The introduction is clear and the paper is pedagogical. The results are timely and important since they make a bridge between the classical and the quantum worlds in the domain of integrable models. This study is also relevant since many experiments simulating the Lieb-Liniger model can access the semi-classical asymptotic regime.

I have a few comments.

  1. As explained in the appendix, the quench action method was used to derive results for a quench from the ground state of the free model (no interaction) to a non vanishing coupling constant, for the Lieb-Liiger model [ref 69]. It might be interesting to make the link with this paper. If the initial temperature chosen for the initial state is reduced, do we recover, with the NLSE, some of the quantum results ? What are the features that can be captured by the NLSE and what are the true quantum features of the final root density function ?

  2. It might be interesting to compare the post-quench root density function (shown in Fig. 8) to a thermal root density function (for instance perform a fit to a thermal distribution). This way the non-Gibbs nature of the final state could be revealed. In the same spirit it might be interesting to know wether and to which extend, the asymptotic values for the moments of the density (shown Fig. 6) differ from those of a thermal state.

  3. I notice two misleading points. (i) p.21, before EQ. 73, I find it strange to change the notation $\lambda$ with $\gamma $ in $T_{\gamma}(t_0;x,y)$ . (ii) In Eq. (77) I guess the matrix element should be $[T_\lambda(t_0;0,L)]_{(1,1)}$ instead of $[ \tau_\lambda(t_0;0,L)]_{(1,1)}$.

Requested changes

None

---

## Round 2 · Referee Report · Anonymous (Referee 2) · 2020-4-14

Strengths

  1. The paper is very well-written and clear in its presentation of the problem.
  2. The paper is very thorough in describing the relevant theoretical background and the tools used to perform the calculations.
  3. In particular, the results presented in section 2 make a very strong case for the formalism presented in the paper.

Weaknesses

  1. Is is a bit hard to define what is the actual main result of the paper, specially in contrast to previous publications.
  2. Since a large portion of the paper is dedicated to revisiting previous theoretical results, the sections containing the original contributions do not stand out as much as they could.

Report

The authors address the problem of the out-of-equilibrium steady states in the classical limit of the interacting Bose gas. They start by outlining the relation between different models (the Lieb-Liniger (LL) model, the Non-Linear Schroedinger Equation (NLSE) and the Sinh-Gordon (ShG) model) and how they map into each other. The authors also review the concept of the Generalised Gibbs Ensemble (GGE) as a tool to describe steady states in the out-of-equilibrium dynamics of integrable systems. In Section 2 the authors then proceed to give a brief description of the LL model and its solution with the Bethe Ansatz approach, as well as its generalisation to the thermodynamic limit in the form of the Thermodynamic Bethe Ansatz. Section 3 holds some of the core results of the paper, where the authors draw the connection between quantities known in the LL model and the corresponding expressions for the NLSE. Here, it becomes clear that finding the root densities for the NLSE is equivalent to obtaining the GGE for this problem and, as a consequence, the values of local observables in out-of-equilibrium dynamics. This sections contains comparisons of the TBA results with the Transfer Matrix method both for one-point functions and probability distributions for particle number density, showing excellent agreement. Section 4 explores the integrability of the NLSE from the standpoint of the Quantum Inverse Scattering Method, and how this formalism allows the authors to obtain the root densities. Section 5 then presents a comparison of the results for the GGE obtained with the tools presented in the previous section with the time-evolution of the NLSE obtained from numerical simulations. The dynamics is obtained after a quench from a non-interacting initial state to a strongly-interacting regime. Again, very good agreement is found both for the density moments as well as for the full count statistics. In section 6 the authors draw the conclusions, with a special focus on the relation between the semiclassical models and their quantum counterparts.

The manuscript is very well-written with an overall very good presentation. The results will certainly be of interest for theorists working with one-dimensional quantum systems, as well as experimentalists dealing with the dynamics of cold atoms.

Here are a few questions and comments for the authors:

  1. In page 3, the authors state that one of the open problems in the quantum regime "is the determination of the steady-state after a quantum quench, which still lacks a general method for generic initial states". Could this passage be improved with the addition of some references that point to this problem?

  2. In page 4, the authors state that "from a physical point of view, the (semi) classical behaviour of a quantum field theory controls the regime when the occupation numbers of the various modes of the field are very high, typically at very high temperatures", when referring to the NLSE as compared to the LL model. How does this relate to the physical interpretation of the Gross-Pitaevskii equation, which is a variation of the NLSE suited for the description of bosonic systems as very low temperature?

  3. It is my understanding that, when comparing results with the TBA approach, the authors refer to the TBA in the semiclassical regime. Perhaps some clarification is in order to avoid confusion with the usual TBA solutions of the LL model, which are also described in the paper.

  4. The interaction parameter is a quantity of great interest from the experimental point of view, given that it can be (usually) tuned with great precision in experimental setups. Throughout the paper, the changes in the interaction strength are described by variations in the chemical potential. In particular, the quench described in Section 5 is stated in terms of the effects of the interaction on the one-point functions. Can some connection be drawn between these quantities and the actual values of the interaction strength in the LL model?

  5. In the conclusions, the authors claim interest in investigating the quantum model by exploring the semiclassical regime. To what extent the results obtained in this regime are able to provide knowledge of the quantum problem? For instance, is it possible to make predictions for the steady states of the LL model with this formalism? What tools would be required for making a direct comparison between the dynamics of the two models?

Requested changes

I have found a few typos throughout the text:

  • Page 3: "...in the Non-Linear Schroedinger."
  • Page 8: "Hence, we consider the bosons to live on a ring with periodic boundary conditions (PBC) since different choices do not affect the physics in the thermodynamic limit." Do the authors mean different choices of size?
  • Page 9: "These two densities are related each other...".
  • Fig. 2 caption: "the partition function of the quantum Lieb Liniger (model?) can..."
  • Fig. 4 caption: "and using te Transfer Matrix approach".

---

## Round 2 · Referee Report · Anonymous (Referee 3) · 2020-4-24

Strengths

  1. Manuscript represents a clear advance in the area.
  2. Complete review of previous related works.

Weaknesses

  1. It contains some long and detailed discussions.
  2. Hard to follow the reading.

Report

The authors submitted a manuscript on out-of-equilibrium properties of a classical integrable system based on the Non-Linear Schrödinger equation. In a sense they develop numerical and analytical calculations to determine the infinitely-many conserved charges of the Generalized Gibbs Ensemble (GGE) in non-relativistic integrable models, to solve the arbitrary quench problem in the classical regime (of high density and energy). This is a fascinating and unsolved problem in the Bose-Einstein condensation and related areas. The subject has acquired further momentum due to a striking progress in new experimental techniques - sometimes called the quantum box - from the world of cold atoms and from new analytic and numerical methods. These astonishing advances have sparked as well the opportunity to explore new phenomena associated with relaxation and equilibration in many-body systems. This is the context where this work is situated.

The manuscript is dedicated to analyse the late-time behaviour of quench protocols, as those given by modifying parameters of the model, to a situation described by semiclassical limits. To this respect, they use the connection of three integrable models, (i) the relativistic quantum sinh-Gordon model, (ii) the quantum Lieb-Liniger (LL) model, and (iii) the classical non-relativistic Non-Linear Schrödinger (NLS) equation. The three models are related at certain semiclassical limits, like velocity of light to infinity or Planck's constant to zero, for example. Extracting crucial information from these relationships they are able to describe exactly the asymptotic steady-state, in terms of the initial conditions, of the out-of-equilibrium dynamics of the classical Non-Linear Schrödinger model, taken as the classical limit of the repulsive Lieb-Liniger model, which is in turn the non-relativistic limit of the quantum sinh-Gordon model.

The organization of the manuscript contemplates, in a topical review style, a thorough discussion of the Hilbert space and the thermodynamics of the Lieb-Liniger model and its projection onto the semiclassical limit, namely, the classical Non-Linear Schrödinger equation. Adopting a coarse-graining approach the exact rapidities of the LL quantum spectrum are described by the Thermodynamic Bethe ansatz (TBA), in terms of a (root) density function. Energy, number of particles and other charges are given in terms of this root density. Having specified the root density is equivalent to determine the GGE, without explicitly know the conserved charges. A technical note is also given to obtain the moments of the root density, through recursive integral equations, which play a principal role in their calculations. After calculating the quantum partition function, through a standard procedure using a path integral approach, the classical limit of NLS is obtained by taking the limit of Planck constant going to zero. They continue in an enforced pace with their derivation until the final result is obtained.

At some point of the manuscript it is hard to be followed. Some analytical and technical descriptions come into the discussion without a continuous development. These include, a discussion on how to deal with the classical limit of the TBA by extracting non-divergent classical root densities from the quantum LL model. There are also a) a derivation of recursive integral equations for the moments of the root density valid for arbitrary GGEs and b) a description of the whole probability distribution of the density operator (full counting statistics). These are non-trivial subjects demanding a concentrated attention from the reader, loosing the perspective of the manuscript. On the other hand, the appendices are too naive in comparison with such a dense manuscript of highly developed analytical derivations. So, the situation is unbalanced.

I would then recommend the authors to find a way to compensate such asymmetry. They should better emphasize what is new and relevant to be put as a definite advance within the manuscript, leaving the details of previous works for the appendices. Or, expand all these details in a more continuous form as to be followed without doing too much gymnastics to get through. One of the authors (G.M.) is already the contributor of the previous results concerning this open problem in the literature. Therefore, it would be easy for them to rewrite such parts of the manuscript to turn it into a pleasant and relevant manuscript.

In conclusion, I think that the manuscript is relevant, clear, globally interesting, and therefore qualifies for publication in SciPost Physics. As an option, my suggestion of a better presentation of the derivation of the analytic results, which I believe is a good exercise on its own, would strongly increase the quality of the paper.

Requested changes

Some typos found: on page 20 a mention to Eq. (61) is given, which does not exist. Some Eqs. mentioned in the captions of Figs. 3, 4 and 6, 7 are in doubt if they are correct. Careful revision of all the captions is demanded. Some overflows of Eqs. (56) and (57) were found on page 19.

---

## Round 3 · Referee Report · Anonymous · 2020-6-22

Report

The authors have addressed the comments and questions contained in the initial report in a satisfactory way.

Requested changes

A few typos:

- In section "The semiclassical limit in the lab"

The semiclassical limit of thermal states and the corrections induced by quantum fluctuations is (are) well controlled...

- In section "TBA in the semiclassical limit"

... which in (the) free system is the mode density...

---

## Round 3 · Referee Report · Anonymous · 2020-6-28

Report

The authors have replied to this referee in a satisfactory way. Since my observations were optional, my recommendation is to accept this manuscript for publication in SciPost Physics.

---

## Round 3 · Author Response

We thank the three referees for their positive comments on our manuscript and for supporting its publication on SciPost Physics. The referees requested some clarifications and put forward some suggestions to further improve our paper: hereafter, we separately reply to each referee describing the changes we made to meet their requests.

Referee 1: We thank the first referee for her/his positive comments. We clarify the issues questioned by the referee using the same numbering used in her/his report. 1. This is indeed an interesting question and comparisons can be made. First, let us point out some caveats: the GS of the non interacting theory in the classical case is nothing else than the constant field phi(x)=constant. No fluctuations neither inhomogeneities are present and the dynamics is easily solved, showing persistent oscillations without any time-relaxation. This pathological limit is due to the absence of statistical fluctuations in the initial state while quantum fluctuations are suppressed in the semiclassical limit. In view of this pathological behavior, we chose not to focus on the dynamics of such an initial state, nevertheless our method can be still applied to compute the GGE root density (even though the system, strictly speaking, does not show any relaxation), which results in a semicircle law. This root density provides sensible predictions for time-averaged quantities. This finding is in agreement with the weakly interacting limit of the quantum result. In the improved version of our manuscript, we explicitly discuss this connection at the end of Appendix D.2. 2. We added a new plot with the requested comparison between GGEs and thermal states. This figure is Fig. 6 in our resubmitted manuscript. Note that the free parameters of the thermal states (beta and mu) are fixed from the inital state. In particular, as it is now discussed in the caption of Fig. 6., beta is unchanged while mu is adjusted to reproduce the initial densities. 3. We thank the referee for pointing out these typos which we promptly corrected.

Referee 2: We thank the referee for appreciating both our results as well as the presentation of our findings. Here, we provide a detailed reply to the referee’s concerns, following the same numeration of the report: 1. Within the quantum case, the determination of the post quench GGEs passes through either computing the charge expectation values (which can be troublesome as we discuss in the manuscript) or through the quench action approach, where one uses the exact computation of certain overlaps among the initial state and the post quench eigenbasis. However, the computation of these overlaps is only possible for very specific initial states. In this respect, a method valid for general states is still lacking. We added the relevant references as the referee suggests. 2. One expects classical physics where statistical fluctuations dominate on quantum ones: in this respect, quantum density matrices that smoothly populate a large part of the Hilbert space can be arguably well approximated by classical physics within a statistical approach. As we extensively discussed, this is the natural conclusion for thermal states with high temperatures. As the referee points out, this is not the only situation where the NLS equation emerges as a description of a quantum model: this is for example the case of the Bose-Einstein condensate. However, the setup is rather different: even though it is true that a single-mode (the condensate) is macroscopically populated, no thermal fluctuations are present, since the system is described by a pure state and not by a density matrix. This results in a purely deterministic dynamics for the classical equation: while an extensive literature has been dedicated to this interesting problem, the relaxation in the GGE sense requires a notion of statistical averaging and thus is not of direct applicability in this case. On the other hand, using classical thermal states to initialize the system, ensures the emergence of a GGE thanks to the averaging on the initial conditions.  3. As the referee correctly understood, we use the quantum TBA as a convenient way to determine the classical TBA, as well as other analytical results presented in the paper. Of course, once the semiclassical limit has been taken, the final result is expressed only in terms of classical objects, thus the classical TBA must be used. In the resubmitted version of our manuscript, we clarify this point at the beginning of the section “TBA in the semiclassical regime”. 4. We use hbar as the small parameter to study the semiclassical limit for the sake of clarity and probably with a more theoretically-oriented taste, but the semiclassical limit can be equivalently reached acting on experimentally-tunable quantities. More precisely, the limit on thermal states can be achieved keeping fixed the chemical potential and taking large temperature and weak coupling, while the product of the latter two is kept fixed. We already pointed out this fact in the previous version of our manuscript, but for the sake of clarity we stress it more in the revised version: at the end of Sec. 3 we wrote a short paragraph explaining how the semiclassical limit can be attained in the lab. We also point out that the quench we explicitly consider in our work is from a non-interacting thermal state to a finite value of the interaction. Therefore, we are changing the interaction and not the chemical potential (which would have led to trivial dynamics) as the referee commented. Provided the fields are correctly normalized, the final classical interaction can always be set equal to 1. At the end of Sec. 5, we now clarify more the connection between the quantum and classical quench, explaining which quantum quenches are approximated by our classical analysis. 5. As we extensively discuss in the manuscript, the physics of the LL model is expected to be classical in the weak coupling/high temperature regime. This has been studied in detail in arXiv:2003.11833 (now included in our reference list) for thermal states, with a systematic inclusion of quantum corrections on the pure classical result. Our work provides a first step in a semiclassical treatment of out-of-equilibrium setups: corrections to the classical approximation are surely a compelling question which we left for future investigations.

We thank the referee for having pointed out some typos which we promptly corrected.

Referee 3: We thank Referee 3 for her/his appreciation of our results, albeit she/he suggests a change in our presentation style. First, we would like to point out that both the other referees explicitly express their appreciation for the presentation (Referee1: “The paper however is well written and can be understood by a large audience. The introduction is clear and the paper is pedagogical.” ; Referee 2: “The manuscript is very well-written with an overall very good presentation.”). We understand that our work contains a certain amount of technical aspects, which are interesting by themselves. Our choice was however to combine such aspects within a discussion of the physical interpretation of both the semiclassical limits involved and the behavior of the out-of-equilibrium dynamics. We chose a more pedagogical exposition that grants accessibility to a broader audience, with a long introduction clarifying the messages of the paper. In the new version, we modify the introduction accommodating for a clearer list of the paper’s content, in such a way the reader can be guided through the presentation to the result she/he is interested in. Those readers who are less interested in the technical details can easily skip some sections and jump directly to the results and the comparison with numerical simulations. On the contrary, an expert reader can quickly move to the technical parts, having already clear in mind the general framework. Let us stress that the results about the density moments and FCS in the classical regime are among our more important results, therefore we believe that they must have a central role in our exposition and they cannot be relegated to appendixes.

We thank the referee for pointing out some typos that have been corrected.

---

## Round 3 · List of Changes

1. A detailed "organization of the paper" section has been added to the introduction to better guide the reader through the main results of the paper.
2. We added the comparison between the analytical results of the quench in the quantum Lieb-Liniger model and our semiclassical approach in Appendix D.3.
3. We compare now the GGEs root density against thermal distributions in Fig. 6.
4. References to the initial-value problem in quantum quenches have been added.
5. The discussion about how to realize the semiclassical limit with experimentally-tunable quantities has been added at the end of Section 4.

---

## Editorial Decision

published